# SatMAE: Pre-training Transformers for Temporal and Multi-Spectral Satellite Imagery

**Yezhen Cong**[*]  **Samar Khanna**[*]  **Chenlin Meng**  **Patrick Liu**
yzcong@stanford.edu  samar.khanna@stanford.edu

**Erik Rozi**  **Yutong He**  **Marshall Burke**  **David B. Lobell**  **Stefano Ermon**

Stanford University

## Abstract

Unsupervised pre-training methods for large vision models have shown to enhance performance on downstream supervised tasks. Developing similar techniques for satellite imagery presents significant opportunities as unlabelled data is plentiful and the inherent temporal and multi-spectral structure provides avenues to further improve existing pre-training strategies. In this paper, we present SatMAE, a pre-training framework for temporal or multi-spectral satellite imagery based on Masked Autoencoder (MAE). To leverage temporal information, we include a temporal embedding along with independently masking image patches across time. In addition, we demonstrate that encoding multi-spectral data as groups of bands with distinct spectral positional encodings is beneficial. Our approach yields strong improvements over previous state-of-the-art techniques, both in terms of supervised learning performance on benchmark datasets (up to ↑ 7%), and transfer learning performance on downstream remote sensing tasks, including land cover classification (up to ↑ 14%) and semantic segmentation. Code and data are available on the project website: https://sustainlab-group.github.io/SatMAE/

## 1  Introduction

In recent years, self-supervised learning techniques have quickly become the norm for pre-training models on large-scale natural image datasets [1, 2, 3, 4, 5, 6, 7, 8], and have demonstrated strong performance on downstream tasks including image classification [3, 4, 9, 10], image segmentation [3, 11], representation learning [12, 13, 14], image compression [12, 15], image reconstruction [1], and image generation [16]. Unlike supervised learning approaches, self-supervised learning techniques do not require human labeling, making them appealing in settings where unlabeled data are abundant but labeled data are scarce, such as remote sensing data (e.g., satellite imagery). While several large-scale satellite image datasets have been carefully curated in the past few years, including Functional Map of the World (fMoW) [17], BigEarthNet [18], xView [19], SpaceNet [20], annotating these datasets requires specialized skills and is more expensive than traditional computer vision datasets. Moreover, automatic analysis of satellite imagery is often needed for tasks with large societal impact such as poverty or crop yield prediction [21, 22, 23, 24, 25, 26, 27, 28, 29, 30], where acquiring large amounts of labeled data through surveys is impossible or prohibitively expensive. This suggests that self-supervised learning approaches for satellite imagery could be especially valuable.

However, existing self-supervised learning approaches [1, 2, 3, 4, 5, 6] are mainly designed for natural images. As opposed to natural images such as ImageNet [31], satellite imagery is usually associated

---

[*]Equal contribution. Order determined via coin flip.

36th Conference on Neural Information Processing Systems (NeurIPS 2022).

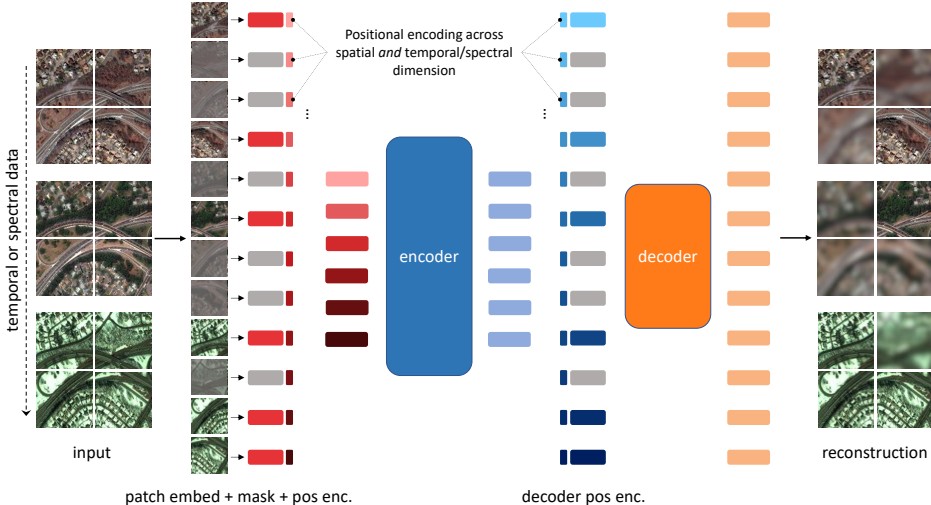

Figure 1: With carefully-designed masking strategies across mutli-spectral and temporal images, and temporal and spectral positional encodings, our SatMAE serves as a powerful SSL vision learner for remote sensing tasks.

with meaningful geographical and temporal information, and can consist of multiple spectral bands representing sensor readings besides visible light (i.e., RGB channels typical in natural images). Depending on the data source, satellite imagery can also vary significantly in resolution [32, 33]. While self-supervised learning methods for satellite imagery exist [34, 35], these approaches cannot learn general representations for both temporal and multi-spectral remote sensing data.

To address this issue we propose **SatMAE**, a self-supervised learning framework based on masked autoencoders (MAEs) [1] which naturally handles temporal and multi-spectral input data. We show that introducing a positional encoding for the temporal/spectral dimension and *independently* masking patches across the temporal/spectral dimension benefits pre-training, allowing the model to learn representations of the data that are more conducive to finetuning. Specifically, our contributions are:

1. We propose a novel method to leverage temporal or multi-spectral information in satellite imagery to improve self-supervised pre-training with masked autoencoders (see 4).

2. We introduce fMoW-Sentinel, a new Sentinel-2 dataset cross-referenced with fMoW, as a benchmark for training models on multi-spectral satellite imagery (see 5.1).

3. We demonstrate the effectiveness of pre-training transformers [36] on satellite imagery, achieving significant improvement over previous state-of-the-art methods on benchmark datasets as well as downstream remote sensing tasks (see 5)

## 2 Related Work

**ML for SITS**    Deep learning has been used for many Satellite Image Time Series (SITS) supervised-learning tasks such as crop-type mapping [29, 28, 37, 38], yield prediction [39, 40], understanding the economy [41, 42, 43, 44], precipitation forcasting [45], and land-cover classification [46, 47, 48, 27]. These works establish the usefulness of tailoring architectures such as LSTMs, self-attention, and transformers to temporal data. However, outside of their specific task, they are often not directly applicable to other remote-sensing datasets.

**SSL for Satellite Imagery**    Self-supervised learning [2, 3, 4, 5, 6] has emerged as a promising approach in remote sensing domains. For instance, [34] and [35] propose incorporating spatially aligned images over time for contrastive self-supervised learning. Despite promising results, these two contrastive learning approaches rely heavily on the quality of positive pairs, which is often hard to control. [49] combines different sensor channels to generate co-located images that serve as positive pairs. [50, 51, 52] apply off-the-shelf contrastive learning algorithms to satellite images. [52] utilizes image inpainting and transformation prediction as additional pretext tasks. [53] leverages geographical knowledge to aid SSL, which, however, can be difficult to obtain as annotations.

**Masked Autoencoder** MAE [1] is a recent powerful self-supervised learning method. Instead of constructing a contrastive objective, it proposes the pretext task of reconstructing masked patches of the input, and largely avoids the need for designing specific data augmentation. Inspired by MAE's state-of-the-art performance on a wide collection of vision benchmarks [1], many follow-up works extend MAE to different data modalities. VideoMAE [54] proposes video tube masking and reconstruction as a pretext task for video analysis. GMAE [55] adapts MAE to the domain of graphs. MultiMAE [56] takes optional inputs of different modalities and accordingly includes other training objectives to facilitate multi-modality learning. However, these works fail to optimally handle temporal and multi-spectral input. VideoMAE requires equally-spaced image frames in the temporal dimension, which is not the case for satellite data given the temporal irregularity and discontinuity in sampling images of a location. In this work, we incorporate temporal and spectral information into a masked autoencoder architecture, and propose a novel self-supervised framework for satellite data.

## 3 Background

**Masked Autoencoder** The MAE is an autoencoder with asymmetrical encoding and decoding stages [1]. It operates on images $I \in \mathbb{R}^{C \times H \times W}$, where $H, W$ are the height and width of the image, respectively, and $C$ is the number of channels. The input image $I$ is resized to a sequence of non-overlapping patches, $S \in \mathbb{R}^{L \times P^2 C}$, where $P$ is the height and width of the patch, and $L = (H/P) \cdot (W/P)$ is the number of patches. Each patch is passed through a patch embedding $f_p : \mathbb{R}^{P^2 C} \mapsto \mathbb{R}^D$ to create a sequence $S' \in \mathbb{R}^{L \times D}$ of embedded patch "tokens". A fraction $p_m$ of the $L$ tokens are masked and only the remaining $(1 - p_m)L$ "visible" patch tokens are fed to the encoder, a Vision Transformer (ViT) [36] with positional embeddings to capture the spatial location of the patch in the image. The decoder is a series of transformer blocks that operates on all $L$ tokens (with positional embeddings added), where the $p_m L$ encoded visible patches are placed in their original sequence position among $(1 - p_m)L$ masked patches represented by a learnable mask token. The decoder outputs a reconstructed image $\hat{I} \in \mathbb{R}^{C \times H \times W}$, which is compared to the original image using the mean-squared error (MSE) loss, computed per-pixel only on the masked patches [1].

**Positional encoding** Positional encoding allows transformers to make their learned representations position-aware. In MAE [1] and in many transformers [57, 58], the positional encoding is:

$$\texttt{Encode}(k, 2i) = \sin \frac{k}{\Omega^{\frac{2i}{d}}}, \ \texttt{Encode}(k, 2i + 1) = \cos \frac{k}{\Omega^{\frac{2i}{d}}} \tag{1}$$

Here, $k$ is the position, $i$ is the index of feature dimension in the encoding, $d$ is the number of possible positions, and $\Omega$ is a large constant (normally set to 10000). In MAE, position is defined as the index of the patch along the x or y axes. Therefore, $k$ ranges from 0 to $H/P$ (or $W/P$). The final encoding is generated by concatenating the encodings of the x and y coordinates.

## 4 Method

In this section, we describe SatMAE with temporal (4.1) and multi-spectral (4.2) satellite images.

### 4.1 Temporal SatMAE

We now consider input tensors $I_T \in \mathbb{R}^{T \times C \times H \times W}$, where $T$ denotes the number of images in a temporal sequence. In video data, $T$ frames are usually equally spaced. However, temporal satellite imagery rarely has images at regular intervals. More commonly, several snapshots, or versions, of a given location are taken at irregular times. The length and sample frequency of these sequences of satellite images vary drastically over years and across different regions.

Naïvely, one could reshape $I_T$ to $I'_T \in \mathbb{R}^{TC \times H \times W}$, effectively concatenating the temporal sequence of images along the spectral (i.e. channel) dimension, and then apply the MAE machinery verbatim. This method poses a few difficulties: (i) the model may be unable to generalise to a temporal ordering different to the one used in pre-training, since it can only understand order through the position of images in the stacked-timeseries (ii) the model cannot reason about the length of time separating two consecutive images in a time sequence, which may be variable when images of a location are

sampled at irregular intervals (iii) the model loses access to temporal fine-grained information in deeper layers, as its only direct exposure to encode temporal information is through the initial patch embedding $f_p$ (iv) the model is not temporally-shift invariant (i.e. the model would need to separately learn to detect the same event in two different segments of a temporal sequence).

To address these challenges and to avoid losing temporal information, we resize the temporal sequence $I_T$ to $S_T \in \mathbb{R}^{L_T \times P_T P^2 C}$, where $L_T = L \cdot (T/P_T) = (H/P) \cdot (W/P) \cdot (T/P_T)$, $P_T$ is the "patch size" in the temporal dimension, and $L$ and $P$ are defined in 3. Prior works using transformers for video data suggest using $P_T = 2$, where each "patch" is a cube of shape $2 \times 16 \times 16$ [54, 59, 60]. Since our data has much shorter temporal sequence lengths [17], we let $P_T = 1$ such that $L_T = L \cdot T$. In order to operate on inputs of any temporal order, we re-use the same patch embedding $f_p : \mathbb{R}^{P^2 C} \mapsto \mathbb{R}^D$ for each image in the time series, giving us an embedded sequence of tokens $S'_T \in \mathbb{R}^{L_T \times D}$.

### 4.1.1 Temporal Encoding

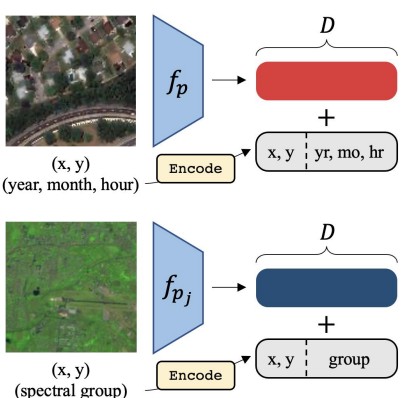

Figure 2: Top: Encoding each temporal patch with a shared patch embedding $f_p$. Bottom: Encoding each spectral patch with a different patch embedding $f_{p_j}$ for each group $j$.

For each embedded token in the $L_T$ length sequence, we need to ensure the model retains information about its spatial and temporal position. As shown in many prior works [34, 35], the timestamp of a satellite image is useful for many pre-training or downstream vision tasks. We propose a temporal encoding scheme compatible with the masked autoencoder architecture by treating the temporal dimension similarly to the positional dimensions (see 3).

The timestamp of a satellite image is represented as "year-month-day-hour-minute-second". Instead of passing the entire numerized timestamp into a feature encoder, we propose only keeping the useful parts. Intuitively, the day, minute, and second should be unrelated to the visual appearance of a region. Thus, including these components in the temporal encoding may not be beneficial, and can even be detrimental. In contrast, a landscape may evolve over years due to weather, geology, and human activity. The month reflects season and climate, and the hour reflects daylight and temperature.

Then, the temporal encoding is formulated as:

$$t_{k,i} = \text{CONCAT}[\text{Encode}(k_{\text{year}}, i), \text{Encode}(k_{\text{month}}, i), \text{Encode}(k_{\text{hour}}, i)] \tag{2}$$

And the final encoding is generated by concatenating the temporal encoding to the positional encoding defined in 3 such that the total length of the encoding is $D$.

### 4.1.2 Masking Strategies

With an additional temporal dimension, masking a subset of the $L_T$ tokens needs to be treated with care. As seen in figure 3, there are different ways to mask a temporal series of satellite images.

**Consistent Masking** Each image is "patchified" separately, but the masked regions are consistent across all images (fig. 3a). This approach is also used in VideoMAE [54], with video input.

**Independent Masking** Each image is "patchified" separately, and masked regions may not be the same across every image. Instead, a fraction $p_m$ of the full sequence of all patch tokens are masked. Another variant is to independently mask the regions of each image, but keep the ratio $p_m$ of masked regions fixed per image. Both variants are equivalent in expectation. Effectively, the model may look at unmasked values of a region that is masked in one image but not in others. This setting may lead to an easier task for video data since the model can "cheat" and exploit temporal redundancy in videos with high framerates [54]. However, we argue that this form of "cheating" is less feasible in temporal satellite imagery, given the strong impact of seasonal variation and changing human activity over periods of time and the much larger time deltas between temporally consecutive images (see fig. 3a).

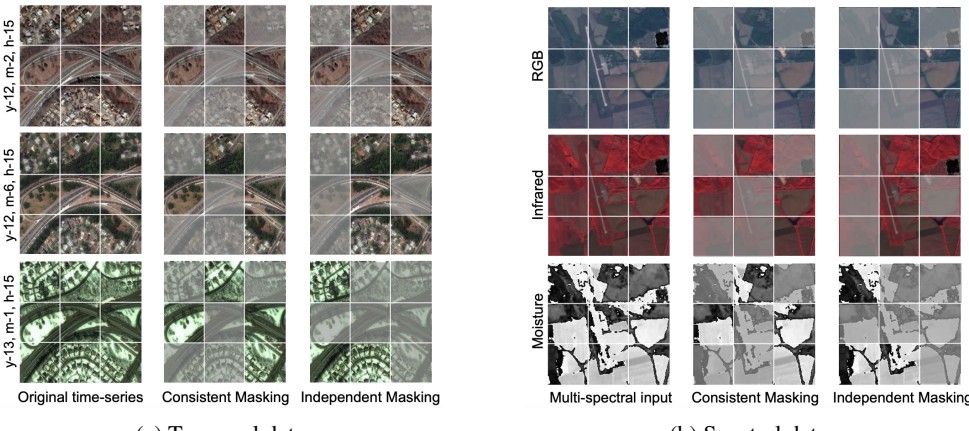

|                     |                      |
|:-------------------:|:--------------------:|
| (a) Temporal data   | (b) Spectral data    |

Figure 3: 3a **Temporal masking**: For images in a timeseries, we can choose to keep a patch fully visible or fully masked across time (*consistent masking*), or independently mask all patches (*independent masking*). In both cases, a fraction $p_m$ patches are masked. Here, $T = 3$, and the leftmost column orders the temporal sequence according to the timestamp features. For example, "y-12, m-12, h-15" is 12 years from the minimum year (2002), the zero-indexed month 2, and the 15th hour of the day; i.e., roughly 2014, March, 15:00. 3b **Spectral Masking**: The same masking strategies are adapted to groups of the 13 spectral bands in Sentinel-2 images.

**Independent Masking + Inconsistent Cropping**   During data pre-processing, we can crop square regions for input inconsistently so that images in the same temporal sequence may be spatially-unaligned. This strategy may help the model learn better representations as it may learn to align images in the sequence across the spatial and temporal dimensions.

## 4.2   Multi-spectral SatMAE

While MAE does operate on images $I \in \mathbb{R}^{C \times H \times W}$, usually $C = 3$ for RGB images. Satellite data, on the other hand, can often have multiple spectral bands. For example, Sentinel-2 imagery has $C = 13$ bands of 10m, 20m and 60m spatial resolution, each of different wavelengths (see A.2.2). Below, we discuss and later experimentally compare various ways to encode spectral information.

**Stack Channels**   The sequence of patches $S \in \mathbb{R}^{L \times P^2 C}$ is embedded to a sequence of tokens $S' \in \mathbb{R}^{L \times D}$, thus treating the multi-band image as is. We denote this method **SatMAE+Stack**.

**Group Channels**   There are limitations to naively stacking the spectral information, especially that a single convolutional patch embedding may be insufficient to fully capture fine-grained information present in multiple bands of different wavelengths and spatial resolution. We would like the model to preserve information about the different bands through the encoding and decoding stages.

To address this limitation, we propose *grouping* subsets of spectral bands. Given $C$ channels, we form $G$ groups $g_1, g_2, \ldots, g_G$ such that $g_1 + g_2 + \cdots + g_G = C$. This is analogous to slicing the image $I$ in the channel dimension, creating images $I_1, \ldots, I_G$, where $I_j \in \mathbb{R}^{g_j \times H \times W}$. We use a separate patch embedding $f_{p_j} : \mathbb{R}^{P^2 g_j} \mapsto \mathbb{R}^D$ for each group $j$, thus allowing the model to best represent each possibly different group of channels as token embeddings. Therefore, each group $j$ is first resized from $I_j \in \mathbb{R}^{g_j \times H \times W}$ to $S_j \in \mathbb{R}^{L \times P^2 g_j}$, and then each patch is embedded with $f_{p_j}$ to produce a sequence of embedded tokens $S'_j \in \mathbb{R}^{L \times D}$. The sequences $S'_1, \ldots, S'_G$ are concatenated to produce the final set of tokens $S' \in \mathbb{R}^{GL \times D}$.

**Spectral Encoding**   Since the tokens in $S'$ correspond to a patch location $(m, n)$ in the input image and a group of channels $g_j$, we include an encoding for the group index $k_g$ similar to 4.1.1

$$g_{k_g, i} = \texttt{Encode}(k_g, i) \tag{3}$$

Note that this encoding simply depends on a user-devised channel grouping, and differs from eq. (2) since additional metadata for the imagery, like its date, is not needed. The final encoding is a

concatenation of the positional $x_{k,i}$, $y_{k,i}$ and the spectral encoding $g_{k,i}$ such that the total dimension is $D$ (see fig. 2). This positional encoding is added to $S'$ before inputting it to the encoder. We denote the combined setting of grouping channels and using a group encoding as **SatMAE+Group**.

**Masking Strategies** We consider *consistent masking* (denoted **SatMAE+Group+CM**) and *independent masking* (**SatMAE+Group+IM**) as defined in section 4.1.2 and as visualized in fig. 3b.

## 5 Experiments

In this section, we first introduce the datasets we considered, including a new multi-spectral remote sensing image dataset for downstream task evaluation (5.1). We then present our results on benchmark datasets (5.2, 5.3, 5.4) and various remote sensing transfer-learning and downstream tasks 5.5. For all experiments, we compare with the current state-of-the-art methods [34, 35] and with supervised learning from scratch using the ViT backbone of SatMAE. In summary, our approach demonstrates strong performance on all the tasks we considered, yielding improvements over previous state-of-the-art techniques by up to 6% on supervised learning benchmarks, and up to 14% on remote sensing transfer-learning downstream remote sensing tasks.

### 5.1 Datasets for Pre-training

**fMoW RGB** Functional Map of the World (fMoW) [17] is a dataset of high-resolution satellite image time series across the world, with a task of classification among 62 categories.

**fMoW Sentinel** We create a new dataset based on the fMoW RGB dataset. We collect all 13 frequency bands provided by Sentinel-2 (B1-12 and B8A) for the original fMoW locations, at some of the same times as fMoW images plus some extra times, for a total of 712,874 training images, 84,939 validation images, and 84,966 test images. More details are included in appendix A.1.

### 5.2 fMoW RGB (non-temporal)

| Method | Backbone | Frozen/Finetune |
|--------|----------|-----------------|
| Sup.* | ResNet50 | -/69.05 |
| Sup.† | ResNet50 | -/69.07 |
| GASSL [34] | ResNet50 | 68.32/71.55 |
| Sup.* | ViT-Large | -/62.48 |
| Sup.† | ViT-Large | -/75.70 |
| Sup.‡ | ViT-Large | -/76.91 |
| SatMAE | ViT-Large | 65.94/**77.84** |

Table 1: Top 1 Accuracy on fMoW classification. **Frozen**: only performing linear classification on frozen features of the pre-trained model. **Finetune**: end-to-end finetuning the whole model. * is training from scratch, and † is using supervised-learning ImageNet weights, and ‡ is SSL MAE ImageNet weights.

In this section, we perform experiments on fMoW single image classification task. Following [34], we report both the performance of linear probing and finetuning setting. Table 1 shows that compared to the previous state-of-the-art self-supervised method using a contrastive momentum encoding approach [34, 3], our SatMAE achieved a 6.29% improvement in top 1 classification accuracy. Interestingly, without SatMAE pre-training the ViT-large model could only reach 62.48% at convergence after 50 epochs of finetuning compared to 69.05% achieved by training a ResNet-50 model from scratch. This is likely because the ViT [36] backbone is harder to finetune from scratch than ResNet50 [61], which makes the pre-trained model more valuable.

### 5.3 fMoW RGB (temporal)

**Main experiments** We perform image-sequence classification on the temporal version of fMoW RGB to evaluate our temporal SatMAE. The temporal fMoW consists of co-located image sequences with a length of 3. As seen in table 2, SatMAE surpasses the previous state-of-the-art by 4.48% and improves the non-temporal result by 2.06% in top 1 classification accuracy. We also outperform UTAE [48], a SITS state-of-the-art, by 18%. We can observe from rows 5-8 that this gain is not from the larger model to handle sequences of data. Naively stacking the image sequences in the channel dimension performs even worse than the non-temporal SatMAE. Again, SatMAE pre-training is crucial for ViT to outperform ResNet50. Training details are in appendix A.3.2.

| Method | Backbone | Top Acc. (1/5) |
|---|---|---|
| Sup.* | ResNet50 | 73.24/- |
| SeCo [35] | ResNet50 | 66.80/- |
| GASSL [34] | ResNet50 | 74.11/- |
| UTAE [48] | U-Net | 61.59/86.45 |
| Sup.* | ViT-Large | 61.89/84.23 |
| SatMAE+Stack | ViT-Large | 75.85/88.68 |
| MAE+Test Aug. | ViT-Large | 78.90/93.31 |
| MAE‖ | ViT-Large | 76.78/92.01 |
| SatMAE | ViT-Large | **81.49/93.26** |

Table 2: Classification results on the temporal fMoW RGB dataset. * means finetuning from scratch. ‖ means copying the input image 3 times instead of using temporal sequences as input. SatMAE+Stack here means stacking the image sequence along the channel space.

| Method | Backbone | Top Acc. (1/5) |
|---|---|---|
| Sup. Learning* | ResNet152 | 49.12/75.73 |
| Sup. Learning‡ | ResNet152 | 54.46/78.99 |
| MoCo-v3 | ViT-Base | 50.45/76.37 |
| MoCo-v3+Group | ViT-Base | 51.33/75.68 |
| SatMAE+Group* | ViT-Large | 53.03/77.14 |
| SatMAE+Group† | ViT-Large | 51.61/77.26 |
| SatMAE+Group‡ | ViT-Large | 47.57/72.26 |
| SatMAE+Group§ | ViT-Large | 49.49/76.30 |
| SatMAE+Stack | ViT-Large | 57.37/81.63 |
| SatMAE+Group+IM | ViT-Large | 59.30/82.81 |
| SatMAE+Group+IM | ViT-Large | **61.48/85.17** |

Table 3: Top 1 & Top 5 Accuracy on the fMoW Sentinel validation set. The different initializations are: * from scratch, † MAE ImageNet weights, ‡ supervised ImageNet weights, § SatMAE fMoW RGB weights. Other rows use fMoW Sentinel for pre-training. The last row includes additional data augmentations (5.4).

| Temp. Enc. | Indep. Mask. | Cons. Crop. | Test Aug. | Top 1 Acc. |
|---|---|---|---|---|
| | ✓ | ✓ | | 78.07 |
| ✓ | | ✓ | | 78.45 |
| ✓ | ✓ | | | 79.90 |
| ✓ | ✓ | ✓ | | 79.69 |
| ✓ | ✓ | ✓ | ✓ | **81.49** |

Table 4: Ablation studies on different components of temporal SatMAE on the temporal fMoW classification task. The first column is whether using temporal encoding, the second is whether using independent masking, the third is whether cropping consistently, and the last one is whether applying test-time augmentation.

| Back. | Group Strat. | Indp. Mask. | Spec. Enc. | Top 1 Acc. |
|---|---|---|---|---|
| Base | X | ✓ | ✓ | 59.11 |
| Large | X | ✓ | | 58.87 |
| Large | X | | ✓ | 57.76 |
| Large | H | ✓ | ✓ | 57.78 |
| Large | R | ✓ | ✓ | 58.76 |
| Large | X | ✓ | ✓ | **59.30** |

Table 5: Ablation studies on spectral SatMAE on fMoW-Sentinel. The first column denotes using ViT-Base or ViT-Large. The second column is the grouping strategy (see 5.4). The third column denotes independent or consistent masking. The last column is whether the spectral group encoding 3 is used.

**Ablation studies**    Table 4 provides a comprehensive ablation study on the components of temporal SatMAE. We see that improved performance is mainly due to the temporal encoding and adopting *independent masking* rather than the consistent masking strategy suggested in VideoMAE [54]. Interestingly, consistent cropping slightly decreases performance, indicating that the model does not rely on perfectly spatially-aligned image sequences. In addition, using test-time augmentations similar to [34] is beneficial. Further ablations on mask ratio $p_m$ and patch size $P$ are in appendix A.4.

### 5.4   fMoW Sentinel (Multi-spectral)

In this section, we pre-train and finetune SatMAE on the image classification task of the fMoW-Sentinel dataset. We pre-train SatMAE+Stack 4.2 and investigate SatMAE+Group+CM 4.1.2 and SatMAE+Group+IM 4.1.2, (see 4.2, 4.2). The full models are then finetuned on the fMoW-Sentinel image classification task. For comparison, we also finetune the ResNet-152 model [61] from scratch and from a supervised ImageNet initialization. We pick the largest model, ResNet-152, for fairer comparison with ViTs. We also include MoCo-v3 [62, 3], a popular SSL method. Given the differences in applying RGB-image augmentations to satellite imagery, we implement two versions: (i) MoCo-v3: we apply all of the same augmentations, except random grayscale and solarize, to create 2 views of the 10-channel image. (ii) MoCo-v3+Group: we split the 10 bands into two groups suggested by [2], and apply augmentations to each to create a positive pair of two 5-channel images.

**Model configuration**    As not all of the 13 Sentinel-2 bands may be useful, in our experiments we drop bands B1, B9 and B10, which correspond to a spatial resolution of 60m. Of the remaining 10 bands, we form three groups: (i) RGB+NIR: B2, B3, B4, B8 (ii) Red Edge: B5, B6, B7, B8A (iii) SWIR: B11, B12. We choose this grouping to ensure each group has bands of the same spatial resolution and similar wavelength (see A.2.2, A.6). Only the last row of table 3 includes additional data augmentations used during finetuning as in [1]. See A.3.3 for pre-training and finetuning details.

| Method | Backbone | Top 1 Acc. |
|---|---|---|
| Sup. (Scratch) | ResNet50 | 54.46 |
| GASSL [34] | ResNet50 | 57.63 |
| Sup. (Scratch) | ViT-Large | 69.65 |
| SatMAE | ViT-Large | **71.77** |

Table 6: NAIP land cover classification results.

| Method | Backbone | mIoU |
|---|---|---|
| Sup. (Scratch) | ResNet50 | 75.57 |
| GASSL [34] | ResNet50 | **78.51** |
| Sup. (Scratch) | ViT-Large | 74.71 |
| SatMAE | ViT-Large | 78.07 |

Table 7: SpaceNet v1 building segmentation results.

| Method | Backbone | Top 1 Acc. |
|---|---|---|
| Sup. (Scratch) | ResNet18 | 63.21 |
| Sup. (IN init.) | ResNet18 | 86.44 |
| GASSL [34] | ResNet18 | 89.51 |
| SeCo [35] | ResNet18 | 93.14 |
| SatMAE* | ViT-Large | 95.74 |
| SatMAE | ViT-Large | 98.94 |
| SatMAE+Group+IM | ViT-Large | **98.98** |

Table 8: EuroSAT land cover classification results. * means we only use the RGB channels of the data.

| Method | Backbone | mAP |
|---|---|---|
| Sup. (Scratch) | ResNet50 | 69.49 |
| Sup. (IN init.) | ResNet50 | 80.04 |
| GASSL [34] | ResNet50 | 80.20 |
| SeCo [35] | ResNet50 | **82.62** |
| Sup. (Scratch) | ViT-Large | 80.07 |
| SatMAE | ViT-Large | 82.13 |

Table 9: BigEarthNet multi-label classification results. Following [35], we use mean Average Precision (mAP) as the metric, and use a newer set of class labels.

**Results** We present results in table 3. Our method SatMAE+Group+IM achieves the highest accuracy, outperforming supervised training from scratch (↑ 6.27%) and ImageNet-initialized backbones (↑ 4.84%). ImageNet initializations may be less useful than in fMoW-RGB given the larger distributional shift to multi-spectral input data. We also note the effectiveness of grouping channels over processing all bands only at the patch embedding level (i.e. SatMAE+Stack).

**Ablation Studies** We investigate the design of SatMAE for multi-spectral data in table 5. For grouping strategy, we implement alternate band groups to test the hypothesis that grouping bands based on wavelength and resolution is beneficial. X represents the band groups in 5.4. H represents splitting the 10 bands into two halves, {(2,3,4,5,6), (7,8,8A,11,12)}. R represents a random split into three groups {(6,5,11,12), (8A,4,8,3), (7,2)}, reflecting the same group sizes as X. As seen, the choice of band groups does influence performance, yielding a gain of about 0.6%. Moreover, ViT-Base performs strongly, suggesting that SatMAE is the reason for improved performance rather than the number of parameters in ViT. Interestingly, *independent masking* performs the best, which prompts the model to "peek" at unmasked band groups to reconstruct the same region in a masked band group.

We also include further experiments on the length of pre-training (see A.3.3), the impact of mask ratio $p_m$ and patch size $P$ (see A.5), and the usefulness of the 13 Sentinel-2 spectral bands (see A.6).

## 5.5 Transfer Learning Experiments

Now, we finetune our pre-trained SatMAE on downstream tasks on remote-sensing datasets, including land cover classification (5.5), multi-label classification (5.5), and building segmentation (5.5). Finetuning details are included in A.7, A.8, A.9, A.10.

**Land Cover Classification** We perform transfer learning experiments on land cover classification using the NAIP and EuroSAT [63] dataset. NAIP consists of RGB+CIR images of 66 land cover classes obtained by the USDA's National Agricultural Imagery Program, which are split into 244,471 training and 55,529 validation images. EuroSAT is a small dataset containing 27,000 13-band satellite images of 10 classes based on Sentinel-2. We follow [35, 64] for the train/val splits on EuroSAT.

Table 6 and table 8 shows the remarkable improvement of our SatMAE over the state-of-the-arts. Although using the ViT-Large backbone already achieved good results, initializing the model with SAT-MAE pre-trained weights further increased the accuracy by 2%-3%.

**Multi-label Classification** We also use the BigEarthNet [18] dataset for multi-label classification, which consists of 13-band Sentinel-2 images of 19 classes in total. There are 354,196 images for training and 118,065 images for validation. Following [35], we use a 10% subset of the train set.

Table 9 shows SatMAE pre-training improves upon the model trained from scratch by over 2%, and achieves comparable results to the state-of-the-art. GASSL and SeCo were actually trained on a larger pre-train dataset (1M Sentinel-2 images v.s. 713k) and with all 13 bands than our fMoW Sentinel. Therefore we expect further improvement when we pre-train SatMAE with more data and for longer.

**Building Segmentation**    In this section, we evaluate SatMAE on the semantic segmentation downstream task of the SpaceNet v1 dataset [20]. The SpaceNet v1 dataset consists of 6940 high resolution satellite images with segmentation masks for buildings, which are divided into train and test sets of 5000 and 1940 images, respectively.

The results in table 7 show that our method achieves a larger performance gain from supervised learning from scratch compared to [34]. The incompatibility of the ViT backbone with PSANet could explain why the baseline performance is not as strong as that of using a ResNet50 backbone.

## 5.6    Visualizing reconstruction quality for SatMAE

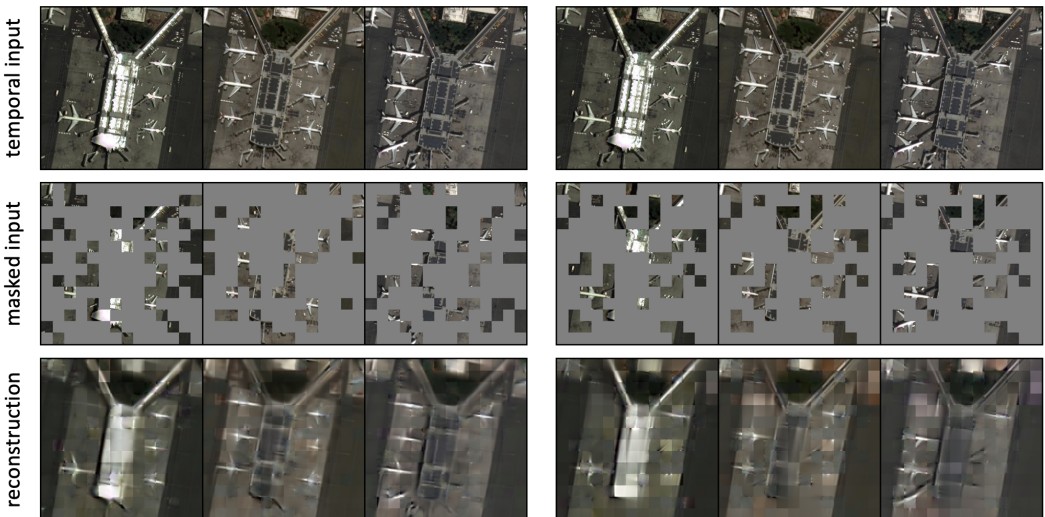

Figure 4: Reconstruction quality of SatMAE+IM (left) vs. SatMAE+CM (right). Further results in appendix C.

We show the visualization of the reconstruction quality of two different SatMAE masking strategies in fig. 4. SatMAE+IM successfully reconstructs all the airplanes even though their number varies across time. In contrast, the SatMAE with Consistent Masking missed some airplanes in the reconstruction.

## 6    Conclusion

In this paper, we propose a new SSL framework based on the MAE architecture [1] tailored to remote-sensing data (satellite imagery). Our novel masking strategy in a joint positional, temporal/spectral space, along with the temporal and spectral encoding, enables our model to handle temporal and multi-spectral satellite images as input and learn useful representations. Experiments on the datasets for pre-training and multiple downstream datasets demonstrate the effectiveness of our pre-trained SatMAE model, outperforming previous state-of-the-art results by large margins.

In the future, it would be useful to design more efficient transformer architectures. While SatMAE has a similar number of parameters for both the temporal and multi-spectral setting as a regular ViT, the increased length of token sequences can strain computational resources. Moreover, it is also worth exploring optimal positional encodings for spectral and temporal data, as well as optimal groups of spectral bands, either by neural-based search methods, or using prior knowledge. Lastly, investigating better architectures for object detection and semantic segmentation using ViTs will be important in generalising SatMAE to further downstream tasks.

## Broader Impact

Accurate measurements of economic, social, and environmental phenomena are key inputs into policy decisions made around the world, but the sparsity of labelled data on many outcomes means that such decisions are often not guided by timely or accurate data. We demonstrate how a pre-training framework could relieve the dependence on labelled data for many downstream tasks that use satellite imagery as input. We hope our SatMAE method will help close the gap between SSL performance on natural imagery and on the more challenging satellite imagery, and prompt further attention from the ML community on the usefulness of SSL in satellite-imagery-related tasks.

Better extraction of information from satellite imagery has profound implications for our ability to measure and understand a broad array of social, economic and environmental phenomena that are critical for decision making. Our approach further amplifies the usefulness of the sparse amount of labelled data that exist on key human outcomes, and could enable rapid and accurate extraction of imagery features relevant for critical downstream tasks, including poverty prediction, infrastructure development, and population estimation. Such information could aid governments in more rapid and data-informed decision making and ultimately bring large societal benefits.

## 7    Acknowledgements

This research is based upon work supported in part by the Office of the Director of National Intelligence (ODNI), Intelligence Advanced Research Projects Activity (IARPA), via 2021-2011000004, HAI, NSF(#1651565), AFOSR (FA95501910024), ARO (W911NF-21-1-0125) and Sloan Fellowship. The views and conclusions contained herein are those of the authors and should not be interpreted as necessarily representing the official policies, either expressed or implied, of ODNI, IARPA, or the U.S. Government. The U.S. Government is authorized to reproduce and distribute reprints for governmental purposes not-withstanding any copyright annotation therein.

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
