# A   Appendix

## A.1   Datasets

**fMoW RGB**   Functional Map of the World (fMoW) [17] is a dataset of high-resolution satellite image time series across the world, with a task of classification among 62 architecture categories such as airport, shipyard, and zoo. fMoW provides RGB images as well as metadata including location, time, sun angles, etc. The license is provided here [2].

Co-located images of different timestamps, or sequences, are provided in fMoW. They are of different length, and around 60% of the samples have length larger than 2. Readers can refer to the fMoW paper [17] for statistics on the distribution of sequence lengths. We construct a temporal version of fMoW by randomly associating every single image with two images of the same location but of different timestamps if possible. For a given spatial location $loc$, we define $T_{loc}$ as the number of temporally distinct snapshots present in the dataset.

**fMoW Sentinel**   We collect a new dataset based on the fMoW RGB dataset. We crop surface reflectance images from the Sentinel-2 (ESA) satellite (courtesy of the U.S. Geological Survey), consisting of 90-day composites of images at the same locations as fMoW images (to reduce the impacts of cloud coverage). At each fMoW datapoint location, we collect a time series of Sentinel-2 images, using the provided geo-coordinate bounding boxes. For locations where all fMoW images are before the Sentinel-2 time range, we discard the location. Otherwise, we collect a composite centered at the same time as each fMoW image within Sentinel-2 time range. Because many fMoW images occur before Sentinel-2, we augment the time series by adding extra images in 6-month intervals that do not have an image in the fMoW dataset.

We collect all 13 frequency bands provided by Sentinel-2 (B1-12 and B8A), at some of the same times as fMoW images plus some extra times, for a total of 712,874 training images, 84,939 validation images, and 84,966 test images. Out of these 155,446 training images, 22,602 validation images, and 22,824 test images occur at the same time as a corresponding fMoW image. The mean height and width of each image is about 45 pixels and 60 pixels, respectively.

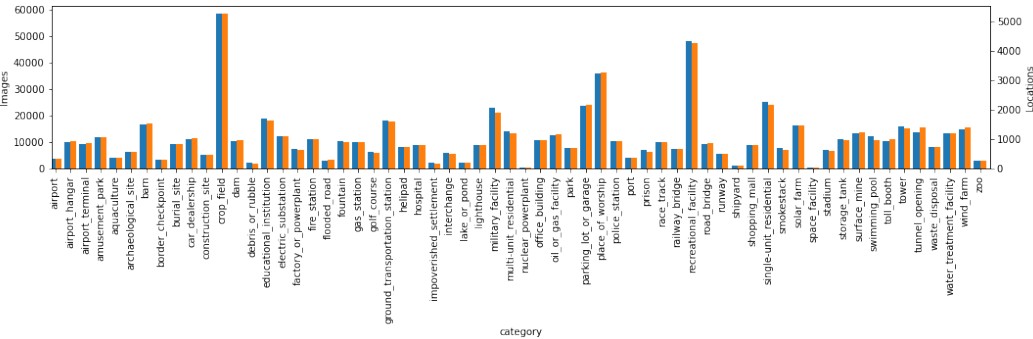

Figure 5: Distribution of images and locations across the categories over the fMoW Sentinel training set.

## A.2   fMoW Sentinel

We provide information about the fMoW-Sentinel dataset, collected using Sentinel-2 [3].

---

[2]fMoW license: `https://github.com/fMoW/dataset/raw/master/LICENSE`

[3]Sentinel-2    license:    `https://scihub.copernicus.eu/twiki/pub/SciHubWebPortal/TermsConditions/Sentinel_Data_Terms_and_Conditions.pdf`

### A.2.1 Geographic Distribution

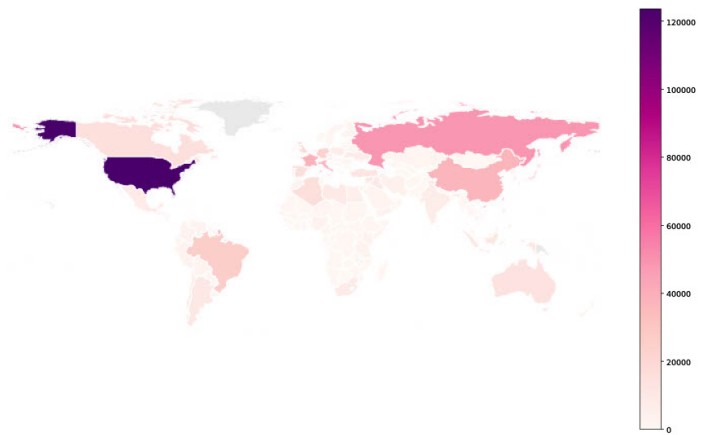

Figure 6: Geographic distribution of fMoW Sentinel images by country

### A.2.2 fMoW Sentinel Bands

| Channel | Resolution | Central wavelength | Mean | Standard deviation |
|---|---|---|---|---|
| B1: Aerosols | 60m | 443nm | 1370.192 | 633.152 |
| B2: Blue | 10m | 490nm | 1184.382 | 650.284 |
| B3: Green | 10m | 560nm | 1120.771 | 965.231 |
| B4: Red | 10m | 665nm | 1136.260 | 948.982 |
| B5: Red Edge 1 | 20m | 705nm | 1263.739 | 1108.067 |
| B6: Red Edge 2 | 20m | 740nm | 1645.403 | 1258.364 |
| B7: Red Edge 3 | 20m | 783nm | 1846.870 | 1233.149 |
| B8: NIR | 10m | 842nm | 1762.595 | 1364.387 |
| B8A: Red Edge 4 | 20m | 865nm | 1972.624 | 3545.66 |
| B9: Water Vapor | 60m | 940nm | 582.726 | 472.380 |
| B10: Cirrus | 60m | 1375nm | 14.771 | 14.311 |
| B11: SWIR 1 | 20m | 1610nm | 1732.164 | 1310.370 |
| B12: SWIR 2 | 20m | 2190nm | 1247.919 | 1087.602 |

Table 10: Mean and standard deviation of pixel values for each channel across the fMoW Sentinel training dataset. Note that channel B10 does not contain bottom-of-atmosphere information, and is no longer accessible on Google Earth Engine. Further details can be found here.

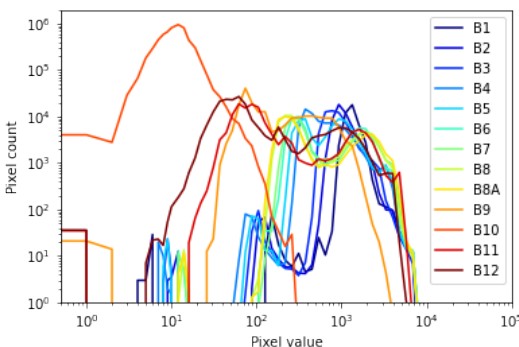

Figure 7: Distribution of pixel counts per band across the fMoW Sentinel training set.

### A.3 Training Details

Here we describe the settings used for pre-training and finetuning our models on fMoW RGB (non-temporal) (A.3.1), fMoW RGB (temporal) (A.3.2), fMoW Sentinel (A.3.3), NAIP (A.7), EuroSAT (A.8), BigEarthNet (A.9), and SpaceNet v1 (A.10).

#### A.3.1 fMoW RGB (non-temporal)

**SatMAE Pre-training** We use ViT-Large [36] as the backbone. The model configuration is the same as in [1], e.g. the input image size is 224 and the patch size is $P = 16$. Since the original image size of fMoW varies greatly, we first resize the image so that the shorter side is 224 pixels and the aspect ratio is maintained, then randomly crop a $224 \times 224$ region from the resized image. We also normalize the image according to the mean and standard deviation calculated on the whole dataset. We use 8 NVIDIA V100 GPUs on Google Cloud to train the model for 800 epochs with a learning rate of $2.4 \times 10^{-3}$ and batch size of 4096. The optimizer and learning rate scheduler are kept the same as in [1].

**SatMAE Finetuning** We load the pre-trained weights below the ViT head and finetune the ViT-Large model in an end-to-end manner. We adopt the same learning rate decay and weight decay strategy during finetuning as in [1]. We apply the same data augmentation during pre-training and additionally use Mixup and Cutmix augmentation. We use 8 NVIDIA V100 GPUs to train the model for 50 epochs with a learning rate of $2 \times 10^{-3}$ and batch size of 512. Other paramters including Mixup coefficients are kept the same as in [1].

#### A.3.2 fMoW RGB (temporal)

**Dataset** We iterate over the dataset in the same way as in non-temporal fMoW, except that we randomly find 2 co-located images with different timestamps (if possible) for every image sample, so every sample becomes a image sequence of length 3. If there are not enough co-located images with different timestamps we simply duplicate the original image. The fMoW dataset train/val split guarantees that co-located images belong to the same split so there is no leakage involved.

**SatMAE Pre-training** We use the same model as above, though the number of input patches triples. To incorporate temporal encoding, the positional encoding of a spatial location of a patch shortens to a $320 + 320 = 640$ dimensional vector, and the temporal encoding is a $384$ dimensional vector, divided equally among the year, month, and hour. We apply constraints on the mask indices to implement different mask strategies. For *independent masking* (4.1.2), we pick the variant where we keep the ratio of masked patches fixed to $p_m = 0.75$ for each image in the sequence. For the consistent cropping option, we first resize the image sequence to the same size and then apply cropping to all three images instead of randomly cropping each image separately. We use 8 NVIDIA V100 GPUs to train the model for 100 epochs with a learning rate of $6 \times 10^{-4}$ and batch size of 1024.

**SatMAE Finetuning** We use 4 or 8 NVIDIA V100 GPUs to train the model for 50 epochs with a learning rate of $5 \times 10^{-4}$ and batch size of 128.

**Test-time Augmentation** Unlike the test-time augmentation used in [34], we average the prediction score of 9 random samples of image sequences for every single image as the final prediction score. To be consistent with previous experiments, we calculate the mean classification accuracy on the whole validation set instead of evaluating on the subset with unique locations. These two metrics give very similar numbers.

**SeCo [35] Pre-training and Finetuning** We use the code from the official repo of SeCo [35], and use ResNet 50 as the backbone. For pre-training, we use 8 NVIDIA v100 GPUs and a batch size of 128, and keep other hyper-parameters and data augmentation the same as in [35]. We pre-train the model for 50 epochs and observe the loss converged. For finetuning, we use 4 NVIDIA v100 GPUs and a batch size of 128, and also keep other hyper-parameters and data augmentation the same as in [35]. We finetune the model for 100 epochs.

**UTAE [48] Training**   We use the code from the official repo of UTAE [48], add an averaging pooling layer to adapt the segmentation network to classification. We use 8 NVIDIA v100 GPUs, a batch size of 128, learning rate of $5 \times 10^{-4}$, and use AdamW optimizer with no weight decay, which we found to be the best performing hyperparameters. We apply data augmentation the same as in SatMAE. We train the model for 50 epochs.

### A.3.3  fMoW Sentinel

We choose the ViT-Large backbone [36] with $D = 1024$. The positional encoding of the spatial location of a patch is a 768 dimensional vector, and the spectral group encoding is a 256 dimensional vector. Given the relatively smaller size of Sentinel-2 imagery, we resize all images to $96 \times 96$ pixels and use a patch size $P = 8$. This results in $L = (96/8)^2 = 144$ patches which are passed to SatMAE+Stack. For SatMAE+Group, since we pick 3 groups of channels, we have $3L = 432$ patches. We did experiment with letting each channel be its own group. However, this resulted in a very large memory footprint with $10L = 1440$ patches and unstable training which would frequently result in NaN loss. We thus decided to group bands in terms of spatial resolution and wavelength similarity (see 5.4). We train and finetune on the entire training set.

**SatMAE Pre-training**   We use 8 NVIDIA v100 GPUs, an effective batch size of 4096, a base learning rate of $10^{-4}$ and the same warmup and half-cyle cosine decay schedule used by [1]. For each image, we use standard normalisation (see statistics in A.2.2), randomly crop 0.2-1.0$\times$ of the area of the image, resize it to $96 \times 96$ pixels, and randomly flip the image horizontally. We use a masking ratio of $p_m = 0.75$, as was found to be optimal in [1]. We pre-train each model for 50 epochs.

**MoCo Pre-training**   We use 8 NVIDIA v100 GPUs, and pick the ViT-Base backbone and an effective batch size of 512 such that the model fits in memory. We pick a base learning rate of $10^{-4}$ and the same warmup and decay schedule used by [62]. For each image, we use standard normalisation (see statistics in A.2.2), randomly crop 0.2-1.0x of the area of the image, resize it to $96 \times 96$ pixels, randomly apply Gaussian blur with $\sigma \in [0.1, 2]$, and randomly flip the image horizontally. We pre-train each model for 50 epochs and use the 50th epoch checkpoint for all subsequent experiments.

**SatMAE Finetuning**   We use 8 NVIDIA v100 GPUs, an effective batch size of 4096, a base learning rate of $10^{-3}$ and a warmup and decay schedule. We use standard normalisation, and resize each image to $96 \times 96$ pixels.

**SatMAE Further Improvements**   We found increased performance of around 2.18% during finetuning (the last row of table 3) using additional data augmentations as in [1]. This configuration is the same as SatMAE+Group+IM except for an effective batch size of 1024, weight decay of 0.05, drop path of 0.1, reprob of 0.25, mixup of 0.8 and cutmix of 1.0. For all rows, we finetune for 30 epochs, but report results on the best validation set Top 1 accuracy achieved.

| Backbone | Pre-train epochs | Top 1 Acc. |
|----------|------------------|------------|
| ViT-B | 200 | 62.65 |
| ViT-L | 50 | 61.48 |
| ViT-L | 200 | **63.84** |

Table 11: Improvements with longer pre-training.

In table 11, we also report results with increased pre-training. Training SatMAE for 200 epochs, as opposed to 50, yields further improvements in the final top 1 accuracy after finetuning for 30 epochs using the configuration described above. We see that a smaller model, using a ViT-Base backbone, can outperform a model using a ViT-Large backbone with longer pre-training. We hypothesize that longer pre-training can prove to be even more beneficial.

### A.4   Impact of masking ratio and patch size on fMoW-RGB-temporal

Here, we investigate the impact of the masking ratio and patch size for a ViT-Large SatMAE on temporal data (with independent masking and consistent cropping, and without the test time augmentation).

We vary the masking ratio $p_m$ to 0.6 and 0.9 from a default of $p_m = 0.75$ and the patch size $P$ to 22, 32 from a default of $P = 16$ (table 12).

| $p_m$ | $P$ | Top 1 Acc. |
|-------|-----|------------|
| 0.6   | 16  | 72.80      |
| 0.9   | 16  | 74.78      |
| 0.75  | 32  | 69.31      |
| 0.75  | 22  | 72.08      |
| 0.75  | 16  | **79.69**  |

Table 12: Ablation on $p_m$ and $P$ on fMoW-RGB-temporal.

We see a significant drop in performance of 7.19% with a smaller masking ratio as expected [1], since a lower masking ratio makes it easier for the model to reconstruct masked patches as it has access to more visible patches. A higher masking ratio of 0.9 may result in a difficult pretext task, as too few of the patches in the image remain visible, which may require longer training. Thus, we find that using $p_m = 0.75$ is roughly optimal.

We also note a drop in performance from using a larger patch size, as the model has access to less granular spatial information from the image. This is in line with other MAE works [1, 54]. However, using a larger patch size is also more computationally efficient, so one must consider the tradeoff in accuracy and computational resources.

## A.5 Impact of masking ratio and patch size on fMoW-Sentinel

| $p_m$ | $P$ | Top 1 Acc. |
|-------|-----|------------|
| 0.6   | 8   | 50.68      |
| 0.9   | 8   | 59.28      |
| 0.75  | 16  | 55.02      |
| 0.75  | 8   | **59.30**  |

Table 13: Ablation on $p_m$ and $P$ on fMoW-Sentinel.

In this section, we investigate the impact of the masking ratio $p_m$ and the patch size $P$ (table 13) for SatMAE on multi-spectral data. We use a ViT-Large backbone and the SatMAE+Group+IM setting as this was our best performing design.

As we expect, a lower masking ratio results in a weak pretext task, as the model is able to easily reconstruct the image given more visible patches, and thus its representations are not as useful. Interestingly, $p_m = 0.9$ doesn't result in a large drop in performance, unlike [1]. This suggests that higher masking ratios may be used for multi-spectral data with independent masking, as it results in fewer tokens during the encoding state which could quicken pre-training.

We see that a larger patch size results in worse performance. This is expected, as a larger patch size provides less granular spatial information to the deeper layers of the model, which may dampen its expressive power. As mentioned above, the loss in accuracy must be considered compared against the gain in training speed. A future direction of research could consider the specific gain in speed and drop in accuracy from granular increases to the patch size for further insight.

## A.6 Choosing important multi-spectral bands

As mentioned in 5.4, all 13 bands of the Sentinel-2 data may not be useful. In our experiments, we drop bands B1, B9, and B10. To correctly identify the utility of each band, one would need to pre-train a model with all bands except the one in question, and then measure the performance after finetuning without that band. However, this is prohibitively expensive in terms of computational resources. Instead, we pre-trained a SatMAE+Stack model with a ViT Base backbone on all 13 Sentinel-2 bands of fMoW Sentinel, and then finetuned the model on the image classification task of fMoW Sentinel using all 13 bands. Using the finetuned model, we ran an ablation masking out subsets of bands with the mean value for those bands and measuring the drop in validation set accuracy. Since the model was trained to rely on information of all 13 bands, a small drop in accuracy from masking out some bands indicates that these bands might not be very useful for the model to perform well.

| Bands Masked | Top 1 Acc. | Top 5 Acc. |
|--------------|------------|------------|
| None         | 57.80      | 80.07      |
| B1           | 45.83      | 69.46      |
| B2, B3, B4   | 17.46      | 35.05      |
| B5, B6, B7   | 13.25      | 35.33      |
| B8, B8A      | 16.03      | 35.36      |
| B9, B10      | 55.06      | 78.15      |
| B11, B12     | 27.83      | 54.41      |

Table 14: An ablation to determine which of the 13 bands are least useful to a SatMAE+Stack model pre-trained and finetuned on all 13 bands of fMoW Sentinel. During evaluation, for each image, the relevant bands are masked with their mean value recorded in table 10 and then passed as is to the finetuned SatMAE+Stack model.

As seen in table 14, we notice the smallest drop in accuracy when masking bands B9 and B10. The drop in accuracy when masking B1 is larger, but could be due to the model relying on potentially unimportant signals from B1 during finetuning. We therefore also drop B1 in our experiments in section 5.4. We note that the RGB and other multi-spectral bands are highly relevant to our model.

### A.7 NAIP Land Cover Classification

We use the finetuning setting as in the fMoW RGB (non-temporal) finetuning experiment (A.3.1).

### A.8 EuroSAT Land Cover Classification

We use the exact finetuning setting as in the fMoW RGB (non-temporal) finetuning experiment for RGB-only input (A.3.1). We also use the exact finetuning setting as in the fMoW-Sentinel finetuning experiment (A.3.3) except for training longer (150 epochs) for multi-spectral (13-band) input. It took the model longer to converge most probably because EuroSAT is comparatively a much smaller dataset. The license[4] is provided in the footnote.

### A.9 BigEarthNet Land Cover Multi-label Classification

We use the exact finetuning setting as in the fMoW-Sentinel finetuning experiment (A.3.3). Since the task is multi-label classification instead of single-label classification, we changed the training objective to multi-label soft margin loss. We use the mean Average Precision metric as provided in [35].The license[5] is provided in the footnote.

### A.10 SpaceNet v1 Building Segmentation

We use PSANet [65] for the binary image segmentation and replace the backbone with ViT-Large. Following [34], we set the learning rate to $1 \times 10^{-3}$ for ViT encoder and to $1 \times 10^{-2}$ for ViT head and PSA module. We train the model for 100 epochs with batch size 128 using an SGD optimizer of momentum 0.9 and weight decay $1 \times 10^{-4}$ and a polynomial learning rate decay scheduler of power 0.9. Also following [34], we resize and crop the input image to $400 \times 400$ for fair comparison. This indicates our model will take more patches per image (625). We use the positional encoding interpolation algorithm provided by [1] to adjust the pre-trained weights. The license[6] is provided in the footnote.

---

[4]EuroSAT license: https://creativecommons.org/licenses/by/4.0/

[5]BigEarthNet license: https://bigearth.net/downloads/documents/License.pdf

[6]SpaceNet v1 license: http://creativecommons.org/licenses/by-sa/4.0/

# B  Societal Impact

Measurements of economic, social, and environmental indicators are critical to policy-making across the world. However, such measurements are constantly lacking, hindering the process of decision making. Instead of using traditional measurements (*e.g.* ground survey), our method exploits abundant, globally-available and frequently-updated unlabelled satellite data. Our model is capable of capturing representations from remote sensing imagery that are beneficial for critical downstream tasks, including poverty prediction, infrastructure development, and population estimation. Governments could make good use of such information in decision making and consequently bring significant societal benefits.

Although the use of satellite imagery could potentially lead to data abuse and privacy violations from malicious actors, we contend that applications of our model trained on publicly available satellite imagery respects privacy and avoids exposing sensitive information. For example, individually identifiable information cannot easily be retrieved from these images. Thus, we believe that the imagery we used does not directly constitute a privacy concern. However, we note that representations learned from SatMAE could potentially suffer from biases if the training data is biased. For instance, SatMAE trained on geographically imbalanced data could bias the model towards certain regions, especially those in Northern America and Europe (see 6). Thus, we advise researchers to be aware of directly applying our SatMAE models to datasets with a geographical distribution different to that of fMoW RGB and fMoW Sentinel. In our code release, we will also specify allowable uses with appropriate licenses.

## B.1  Carbon Footprint

We include a brief analysis of the carbon footprint of training the model below.

Our experiments were mainly conducted using Google Cloud Platform (GCP) in region us-central1, which has a carbon efficiency of 0.57 kg $CO_2$ eq. per kWh. For a model pre-trained and finetuned on fMoW RGB (temporal) dataset, a cumulative of 960 hours of computation was required on hardware of type Tesla V100-SXM2-16GB (TDP of 250W). Total emissions are estimated to be 136.8 kg $CO_2$ eq. of which 100 percent was directly offset by the cloud provider. Estimations were conducted using the Machine Learning Impact calculator presented in [66]. For a model pre-trained and finetuned on fMoW Sentinel dataset, total emissions are estimated to be 109.44 kg $CO_2$ eq. We list a table for the rough estimations in table 15.

| Experiment Setting | Dataset | GPU hours | Carbon Footprint (kg $CO_2$ eq.) |
|---|---|---|---|
| Pre-training | fMoW RGB temporal | 768 | 109.44 |
| Finetuning | fMoW RGB temporal | 192 | 27.36 |
| Pre-training | fMoW Sentinel | 576 | 82.08 |
| Finetuning | fMoW Sentinel | 192 | 27.36 |
| Finetuning | NAIP | 30 | 4.27 |
| Finetuning | EuroSAT | 4 | 0.57 |
| Finetuning | SpaceNet | 50 | 7.12 |
| Finetuning | BigEarthNet | 16 | 2.28 |

Table 15: The estimated carbon footprint of pre-training and finetuning SatMAE on these datasets. The GPU hours are measured on 8 NVIDIA v100 GPUs in the us-central1 region on GCP.

# C Visualizations

In this section, we include visualisations in the temporal (C.1) and multi-spectral settings (C.2).

## C.1 Temporal SatMAE

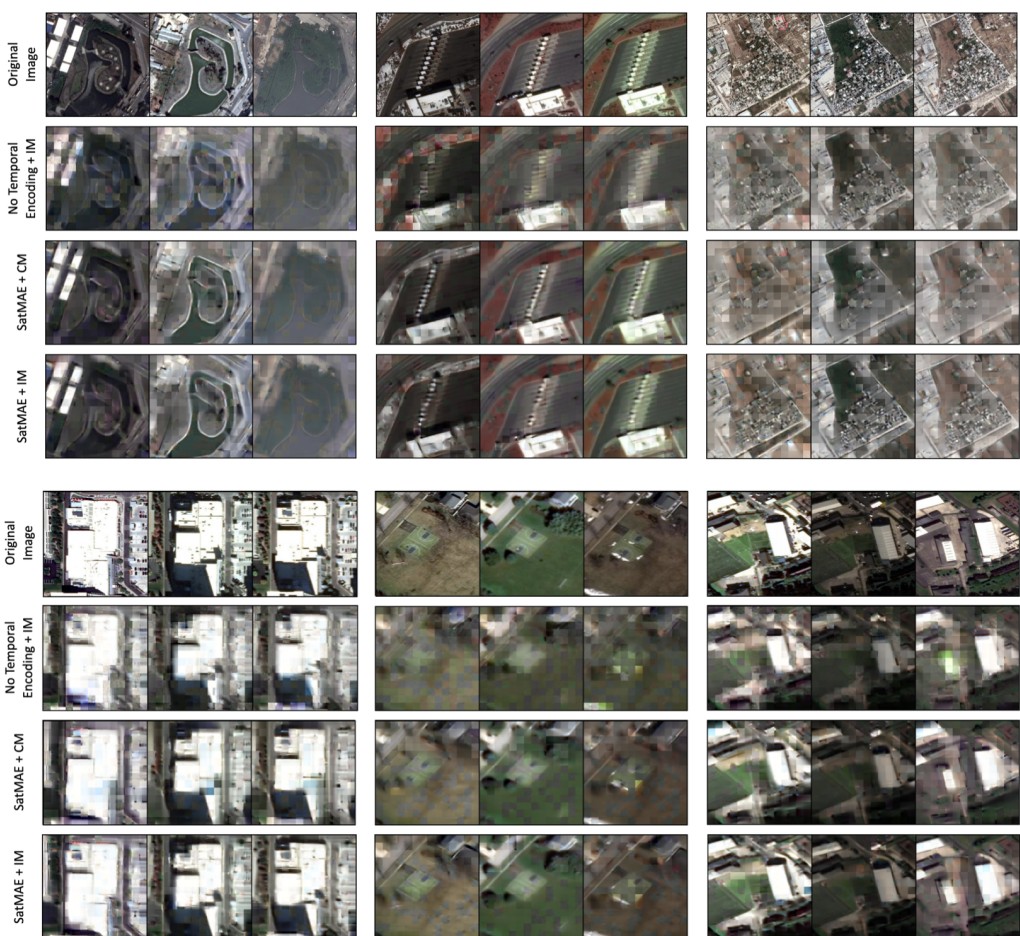

Figure 8: More visualization examples of the reconstruction quality of image sequences from the fMoW temporal dataset across multiple settings, including using no temporal encoding + IM (Independent Masking), default + CM, default + IM.

As shown in fig. 8, SatMAE+IM achieved relatively satisfying reconstruction quality. Without the temporal encoding, the patches across all three images cannot be distinguished, and we observe a mixture of the three images in the reconstruction outcome of the second column. As explained earlier, using independent masking can allow SatMAE to reconstruct an image in the time series using information from other temporal patches. Our experiments show that this helps SatMAE learn better representations for satellite imagery.

## C.2 Spectral SatMAE

We also visualize the in-painting quality for different multi-spectral settings, including Sat-MAE+Group+IM, SatMAE+Group+CM (4.2, 4.2), and SatMAE+Stack (4.2) in fig. 9 and fig. 10.

We can see a clear improvement in the quality of reconstruction under SatMAE+Group+IM compared to SatMAE+Group+CM and SatMAE+Stack. Independent masking results in sharper reconstructions, whereas the results from consistent masking and stacking the channels are much fuzzier. We also

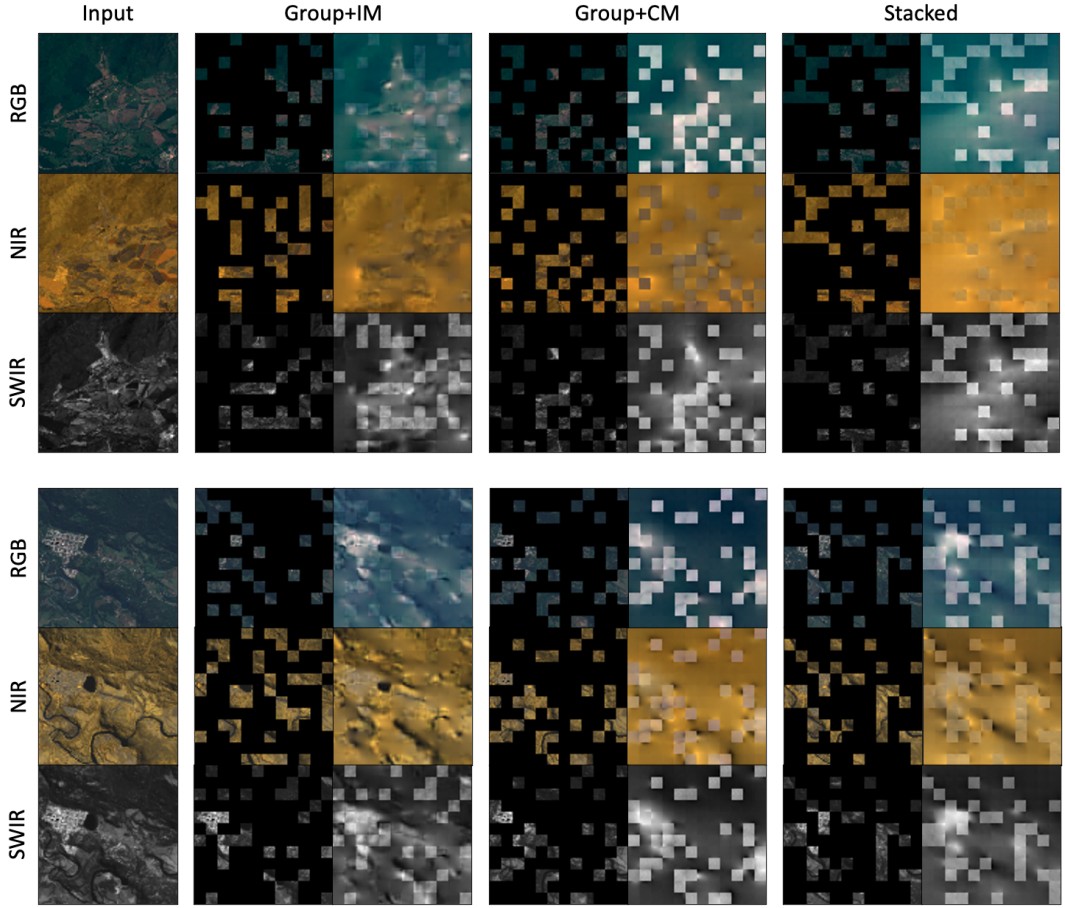

Figure 9: Visualizations of SatMAE in-painting under different settings. RGB represents bands B4, B3, B2, from group B2, B3, B4, B8. NIR represents bands B7, B6, B5, from group B5, B6, B7, B8A. SWIR represents bands B11 in grayscale, from group B11, B12. For each method, we show the input band group masked and reconstructed side-by-side. We note that the reconstruction for visible patches is worse than for the masked patches, since no loss is computed on visible patches. Both halves represent multi-spectral images of airports.

note that the model is able to learn correlations between bands; in the top-half of fig. 9 for the SWIR band group, even though the bottom right corner of the image is masked, SatMAE+Group+IM is able to reconstruct the bright spot based on information from the other band groups.

We hypothesize that further improvements in reconstruction quality and learned representations can be achieved with longer pre-training.

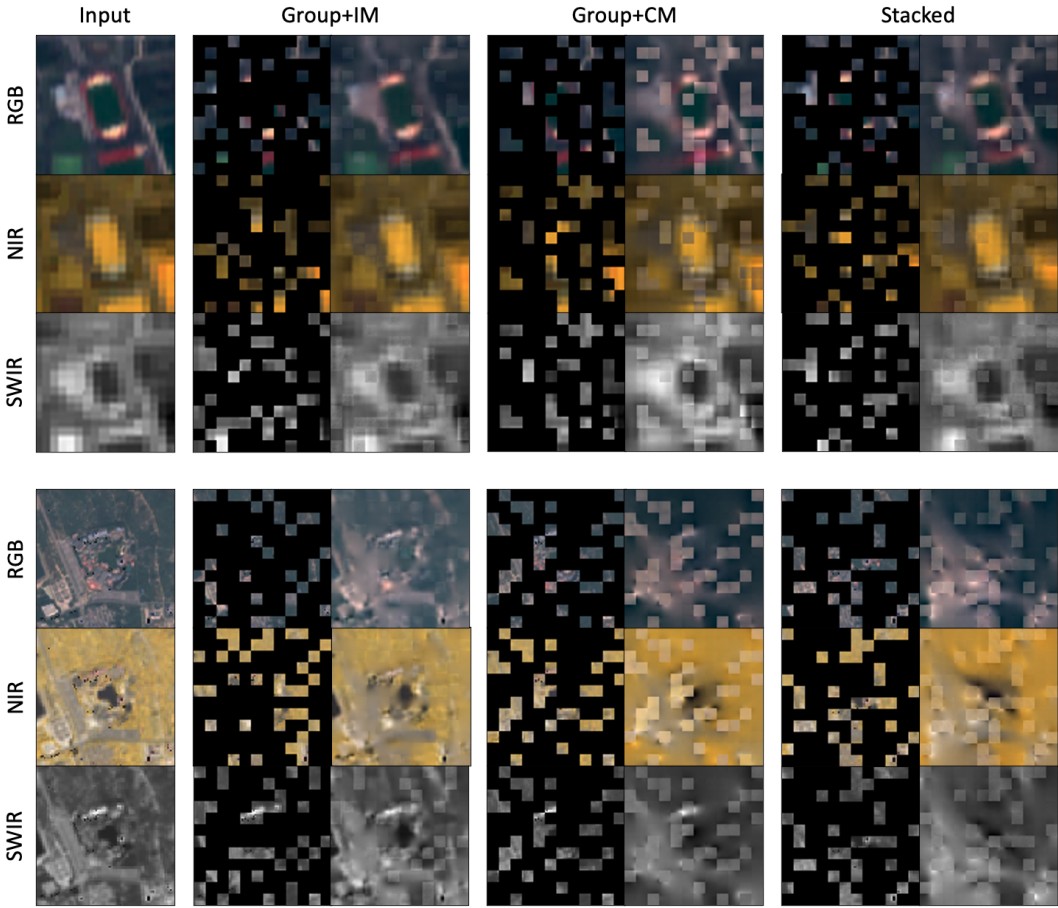

Figure 10: Further visualizations of SatMAE in-painting. See fig. 9 for details on band groups. The top half represents a multi-spectral image of a recreational facility and the bottom half is of an amusement park.