# OpenReview forum: "SatMAE: Pre-training Transformers for Temporal and Multi-Spectral Satellite Imagery"
_NeurIPS.cc/2022/Conference — NeurIPS 2022 Accept_

### Official Review · Reviewer_vWmP · 2022-06-29

**Rating:** 6
**Confidence:** 4
**Soundness:** 3 good
**Presentation:** 2 fair
**Contribution:** 3 good

**Summary:**

The authors present a pretraining method for satellite images based on masked autoencoders. They perform several experiments on different datasets and tasks, and propose a new dataset combining Sentinel 2 time-series and existing land use labels. The proposed method improves the results of their backbone network.

**Questions:**

- please explain why channel grouping is important in spectral space, and comment on the results
- please compare the results with actually dedicated architecture, not spactro-temporal stacked ResNets
- please give additional details on the fine-tuning

**Strengths And Weaknesses:**

Strengths:

+ multiple experiments on several datasets and tasks
+ Curated a new dataset from existing data/annotations
+ An interesting to setting not too well explored in ML/CV yet
+ Use of modern tools not too explored on remote sensing data

Weaknesses:

- method and experimental settings are somewhat unclear: what is used for pretraining and fine-tuning, etc
- some design choices are not well motivated (eg. spectral channel grouping)
- results show that many of the ideas presented have negative effects on the performance, yet this is not addressed
- no ablation study on the design of the temporal position encoding
- the authors completely ignore the entire corpus of SITS classification. Claims of state of the art when the baseline is spectral-temporal concatenation are not very convincing

Overall:
a high-effort paper with many experiments on an interesting subject with overall promising results. However, the article completely ignores the literature on SITS, and some design choices seem actually to decrease the performance.

Missing Refs: all these papers present stronger baselines for SITS analysis. Claim of SOTA should include running some of these appraoches, or explanation why they cannot be applied here:

Rustowicz etal "Semantic segmentation of crop type in africa: A novel dataset and analysis of deep learning methods." CVPR Workshops, 2019

Martinez etal "Fully convolutional recurrent networks for multidate crop recognition from multitemporal image sequences." ISPRS Journal of Photogrammetry and Remote Sensing, 2021.

Rußwurm etal. "Convolutional LSTMs for cloud-robust segmentation of remote sensing imagery." NeurIPS Workshops, 2018.

Garnot etal "Panoptic segmentation of satellite image time series with convolutional temporal attention networks." ICCV, 2021

Shi etal "Convolutional LSTM network: A machine learning approach for precipitation nowcasting." In NeurIPS, 2015.


Details:

Fig1 is unclear and hard to read without legend / detailed caption
In particular, are the features greyed out and not the input patches?

Do not mention nor cite the large body of work on SITS, eg:

It is unclear how the temporal patches deal with uneven temporal samplings.
Motivation for spectral grouping?

Why would an MLP not be expressive enough to analyze the correlated spectral information. 13 bands is not that many, and it is standard practice.

Notation: j used for spatial and spectral indices...

No equation number makes it hard to refer to precise

linesL188 not clear at all, i j are spatial indices; how do they relate to spectral indices?

L190: what is i here?

Flow Sentinel: size of images?

Table2: resnet 50 is a very weak baseline for SITS classification.

Use dedicated networksTab3,7: Group works worse than stacking...

Tab4: temporal encoding worsen prediction... Yet the authors say the opposite

L234The experimental setting is very unclear: how much data are used for fine-tuning? All of it?

---

> ### Author Response · Authors · 2022-08-02
> **Response to Reviewer vWmP**
>
> Thank you for your thoughtful feedback, recognition of SatMAE’s contributions, and suggestions for improvement. Your suggestions greatly helped us improve our analysis of SatMAE. You may find responses to your questions below.
>
> **Q**: Method and experimental settings.
> **A**: We mention this in the supplementary material in section A3.
>
> **Q**: Motivation of design choices  (eg. spectral channel grouping)
> **A**: In our paper, we provided reasoning for the grouping: “... to ensure each group has bands of the same spatial resolution as well as similar wavelength”
> In the appendix A.3.4 we mention details of an experiment where we drop certain sentinel bands from a model trained on all 13 bands and measure the drop in inference performance.
> In our revision, we have also included further experiments on the impact of different band grouping strategies on performance in table 5. While our original design performs the strongest, all designs demonstrate the benefits of grouping channels over the baseline of SatMAE+Stack. As we stated in the original submission,
> “We would like the model to preserve information about the different bands through the encoding and decoding stages”
> Thus, our design choices demonstrate that grouping the bands is beneficial to the model’s performance.
>
> **Q**: some ideas presented may have negative effects on the performance
> **A**: We believe you may be referring to the drop in top 5 accuracy in certain ablations. We emphasize that the primary metric of concern is top 1 accuracy, and top 5 accuracy has been provided simply for completeness. In all our experiments, our design choices increase top 1 accuracy. As top 1 accuracy is the metric of concern for performance on the benchmark datasets as well as any downstream tasks, we believe that these drops are of little significance. It would be an interesting research direction to explore the relations in gains and drops between the two metrics. To be specific, the consistent masking option serves as a negative comparison for the independent masking option, which is the design choice we actually adopt. With or without inconsistent cropping shows little difference, and it is only applicable to non-square input images, which is the case of fMoW but not general.
>
> **Q:** Ablation study on the design of the temporal position encoding
> **A:** We have included an ablation on whether the temporal position encoding is encoded or not in Table 4. We also provide an explanation of the motivation behind the temporal encoding in section 4.1.1. We pick this design to simply encode the relevant temporal information for the model. Moreover, there are a combinatorial number of ways to encode the timestamp differently, and we decided instead to pick a design that makes intuitive sense, directing our limited computational resources towards other ablations in the design of the model.
>
> **Q**: corpus of SITS classification.
> **A**: We have included some of the relevant works in our revision. Many of these works design model architectures for supervised learning on specific downstream tasks, eg: crop-type mapping. We propose SatMAE as a self-supervised learning method for representation learning, and therefore primarily compared our approach with other self-supervised learning methods for satellite imagery. Our claim of state-of-the-art performance is in terms of results on benchmark datasets such as fMoW and on downstream tasks. Moreover, many of the recent works suggest that transformers are particularly suited for time-series classification. Our backbone uses such a model, and is robust to both low-resolution (Sentinel) and high resolution (fMoW) satellite imagery.
>
> **Q**: Clarification on Fig1.
> **A**: The tokens of the input patches are masked and colored gray. As described in section 3, only the visible (non-gray) tokens are passed to the encoder. All tokens are passed to the decoder.
>
> **Q**: dealing with uneven temporal samplings
> **A**: The temporal encoding includes the year, month, and day. This provides the model with information about unevenly sampled images since the gaps between subsequent embeddings will be longer/shorter
>
> **Q**:  MLP is sufficient to analyze the correlated spectral information.
> **A**: The method is intended to be flexible. With multiple channels, it may be beneficial to propagate the spectral granularity to layers deeper in the transformer network rather than fuse that information at once in the beginning, as is done in SatMAE+Stack. As we demonstrate, grouping the bands outperforms treating the 13 bands as a regular image (Table 3, SatMAE+Group+IM vs SatMAE+Stack).
>
> **Q**: Clarification on notation and writing
> **A**: We have revised some parts of Section 4 to use clearer notation. We have also included the average height and width of Sentinel-2 images in the appendix (A.1)

---

> ### Author Response · Authors · 2022-08-02
> **Response to Reviewer vWmP (Contd.)**
>
> **Q**: ResNet50 as a baseline for SITS
> **A**: Many prior works [1, 2] use ResNet 50 as backbones for their SSL methods and have achieved SOTA performance on the benchmark datasets we consider. We compare SatMAE with these architectures as we intend to show the advantages of our SSL method over prior work in unsupervised representation learning for satellite data. We also compare with stronger baselines such as ViT, and demonstrate a marked improvement. As stated in [3], self-attention using transformers is one of the SOTA methods for SITS. [4] also demonstrates the strength of ViT over convolutional approaches. We show the superiority of our method over these baselines in our experiments since we outperform ViT (tables 2, 3).

---

> > ### Comment · Reviewer_vWmP · 2022-08-06
> > **Temporal vs Spatial encoder**
> >
> > There seems to be a confusion between spatial and temporal encoders. [3] is a temporal encoder, and is not in any way comparable to ViT. Outperforming ViT does not prove that stacking temporal channels is a better strategy than using an ad-hoc network to process the temporal dimension.
> >
> > And resnet 50 is not a SOTA network for SITS (eg sentinel), only for single images.

---

> > > ### Author Response · Authors · 2022-08-08
> > > **Response to follow up comment on SITS baselines**
> > >
> > > Thank you for your follow up! Sorry about the misunderstanding, we have included an experiment to run UTAE [1] on fMoW-RGB-temporal for additional comparison. As this method outperforms most other architectures for SITS, including Conv-LSTMs and 3D-UNets, we hope that this serves as a sufficient baseline against which to compare our method on SITS. Since this paper devises this model for segmentation, we add an average pooling layer in the final output for the fMoW-RGB-temporal classification task.
> > >
> > > |  Method   |   Backbone  |   Top 1 Acc.  |
> > > | ---- | ---- | ---- |
> > > |  UTAE [1] |  U-Net    |   61.59   |
> > > | SatMAE (without test aug.)   | ViT         |   79.69
> > >
> > > As seen in this result (which we add to our revision in table 2), the model severely underperforms compared to SatMAE (at ~79% without test-time augmentation) on this dataset. We experimented with a few different hyperparameters for UTAE, and found that learning rate=5e-4, batch size=128, epoch=50 yielded the highest validation accuracy. We train the full UTAE model from scratch.
> > >
> > > Also, we would like to mention that a lot of other SITS models are applied to tasks different to the ones we consider. [2, 3] use models that do not operate on input images, but rather parcels. [2] operates on Sentinel-1 parcels, which are not applicable to the high resolution images in fMoW-RGB-temporal. [3] uses Sentinel-2 pixels, averaged over field parcels, and processes tensors of shape TxD which are not images. Therefore, their models are not directly applicable to our task.
> > >
> > > We also wanted to clarify that the results we display for fMoW-Sentinel are for the single input image classification task, so this task is not directly comparable to SITS. fMoW-RGB-temporal is the benchmark dataset that we use for temporal classification.
> > >
> > > Lastly, we wanted to emphasize that our method is a self-supervised learning pretraining method that can be used to train ViTs for a variety of downstream satellite imagery tasks. We don’t aim to beat every baseline on the downstream task, and instead hope to show robust and strong performance with the flexibility of our approach. On many benchmark datasets, eg: fMoW-RGB, fMoW-RGB-temporal, NAIP, EuroSAT, we beat the state of the art SSL pre-training methods. We aim to promote SatMAE as a very useful pre-training algorithm for temporal/multi-spectral satellite imagery.

---

> > > > ### Comment · Reviewer_vWmP · 2022-08-09
> > > > **Ackowledgment of response**
> > > >
> > > > This fixes my main concern about the paper. While the novelty and the improvements remain rather moderate, the ideas explored are important and the experiments well-conducted.
> > > >
> > > > I updated my rating accordingly.
> > > >
> > > > Thanks to the author for the hard work they put into this rebuttal.

---

> > > > > ### Author Response · Authors · 2022-08-09
> > > > > **Thank you for your response**
> > > > >
> > > > > Thank you very much for considering our revisions and updating your evaluation of our paper. Your suggestions are very constructive and helpful for improving the quality of our work and the paper.

---

> > > ### Author Response · Authors · 2022-08-08
> > > **References accompanying response to follow up comment on SITS baselines**
> > >
> > > [1] Garnot, V. S. F., & Landrieu, L. (2021). Panoptic segmentation of satellite image time series with convolutional temporal attention networks. In Proceedings of the IEEE/CVF International Conference on Computer Vision (pp. 4872-4881).
> > >
> > > [2] Martinez, J. A. C., La Rosa, L. E. C., Feitosa, R. Q., Sanches, I. D. A., & Happ, P. N. (2021). Fully convolutional recurrent networks for multidate crop recognition from multitemporal image sequences. ISPRS Journal of Photogrammetry and Remote Sensing, 171, 188-201.
> > >
> > > [3] Rußwurm, M., & Körner, M. (2020). Self-attention for raw optical satellite time series classification. ISPRS journal of photogrammetry and remote sensing, 169, 421-435.

---

> ### Author Response · Authors · 2022-08-02
> **References Accompanying Our Response to Reviewer vWmP**
>
> [1] Ayush, K., Uzkent, B., Meng, C., Tanmay, K., Burke, M., Lobell, D., & Ermon, S. (2021). Geography-aware self-supervised learning. In Proceedings of the IEEE/CVF International Conference on Computer Vision (pp. 10181-10190).
>
> [2] Manas, O., Lacoste, A., Giró-i-Nieto, X., Vazquez, D., & Rodriguez, P. (2021). Seasonal contrast: Unsupervised pre-training from uncurated remote sensing data. In Proceedings of the IEEE/CVF International Conference on Computer Vision (pp. 9414-9423).
>
> [3] Rußwurm, M., & Körner, M. (2020). Self-attention for raw optical satellite time series classification. ISPRS journal of photogrammetry and remote sensing, 169, 421-435.
>
> [4] Kaselimi, M., Voulodimos, A., Daskalopoulos, I., Doulamis, N., & Doulamis, A. (2022). A Vision Transformer Model for Convolution-Free Multilabel Classification of Satellite Imagery in Deforestation Monitoring. IEEE Transactions on Neural Networks and Learning Systems.

---

### Official Review · Reviewer_Wua7 · 2022-07-07

**Rating:** 6
**Confidence:** 4
**Soundness:** 3 good
**Presentation:** 3 good
**Contribution:** 3 good

**Summary:**

The paper introduces a new masked autoencoder, namely SatMAE, to pretrain for temporal or multi-spectral satellite imagery. The method aims to reconstruct the masked parts of the input under a high masking rate. More specifically, the paper presents several simple yet useful masking strategies and temporal/spectral encoding schemes based on the properties of remote sensing images. The proposed SatMAE is systematically on several challenging benchmark datasets, showing state-of-the-art results. The authors also create a new dataset based on the fMoW RGB dataset. Overall, the proposed architecture is simple but seems effective for RS data and the paper is well-presented.

**Questions:**

1. As the most important baseline,  why didn't the standard MAE be compared in the experiment? Especially, in the ablation study,  the standard MAE should be compared as the baseline in Table 4. This is important to indicate the effectiveness of the proposed strategies.
2. To indicate the superiority of the proposed method, different closely-related SSL architectures could be compared, for example, SimCLR, MoCo, DINO, and recent masking prediction models (e.g. MaskFeat).
3. The temporal/spectral encoding is unclear. More explanation is needed. For example, what are the differences between positional embedding and temporal/spectral encoding, and how are they encoded with patch embedding?

**Ethics Review Area:**

["I don’t know"]

**Limitations:**

The authors provided a suitable discussion about the broader impact of the proposed method.

**Strengths And Weaknesses:**

Strengths:
1. The paper presents a new self-supervised pertaining architecture for RS data. The proposed method is simple but effective.
2. The performance is evaluated on challenging datasets, and the experiment results are sufficient and promising.
3. A new dataset is created.

Weaknesses:
The core contribution of the paper is to extend MAE to temporal or multi-spectral satellite imagery and design several new masking and positional encoding strategies accordingly. From this perspective, the novelty of the proposed method is a bit incremental.

---

> ### Author Response · Authors · 2022-08-02
> **Response to Reviewer Wua7**
>
> Thank you for your thoughtful feedback and recognition of SatMAE’s novelty, soundness, and effectiveness. We also appreciate your observant suggestions on building more solid baselines and clarifying the design details of our approach. You may find responses to your questions below.
>
> **Q:** *Comparison with the standard MAE.*
>
> **A:** This experiment was performed in our original submission. In our tables, this is represented as the SatMAE+Stack. As the input size in the temporal or multi-spectral setting is N*H*W, where N is not 3 as for single RGB images, standard MAE [1] must change the input channel size parameter in order to handle temporal or multi-spectral data. To be more specific, we adopted the whole config from MAE except for the number of channels (which would be 9 for temporal, and 10 for multi-spectral setting). We think this is the most straightforward definition of standard MAE extended to temporal and multi-spectral data, and we also tried other variants, see Table. 2 and Table. 3.
>
> For the fMoW RGB temporal setting, since the temporal dataset is only leveraging timestamps additional to existing images and labels, we can also compare the MAE results on non-temporal fMoW (last row of Table. 1 in revised paper), 77.84%, with our method designed for temporal data, 81.49%. In both comparisons, there is a large gap between the MAE baseline and our method.
>
> **Q:** *Comparison with other SSL architecture.*
>
> **A:** We have included a comparison with MoCo-V3 [2] using ViT-base (not using ViT-Large due to computational issues) for fMoW Sentinel (third and fourth row in Table. 3 of our revised paper), and we also included SeCo [3] (second row in Table. 2 of the revised paper), which is a state-of-the-art SSL technique for temporal remote sensing data, for fMoW-temporal, in our revised paper. There is a wide gap between theirs and our method. Unfortunately, due to computational constraints, we were unable to run all the other baselines ourselves. However, in Table. 1, Table. 2, Table. 3, we outperform GASSL [4], SeCo and MAE, which outperform the related SSL methods, including MoCo-V3, DINO [5], and SimCLR [6] (e.g. MAE outperform DINO in their paper). By transitivity, we believe this shows the superiority of our approach.
>
> **Q:**  *Explanation on temporal/spectral encoding.*
>
> **A:** The difference is in the fact that MAE only includes positional encoding, we also incorporate temporal/spectral. As described in section 4.1.1, the positional encoding is concatenated with the temporal/spectral encoding. Then they are element-wise added to the patch embedding, which follows the convention of MAE [1] (please refer to the MAE paper). One way to understand the temporal/spectral encoding is by thinking of it as another axis additional to the x-axis and y-axis of the 2D RGB image. One different thing about temporal encoding, though, is that it tackles the irregularity along the temporal axis by encoding year, month, and hour, respectively.

---

> ### Author Response · Authors · 2022-08-02
> **References Accompanying Our Response to Reviewer Wua7**
>
> **References:**
>
> [1] He, K., Chen, X., Xie, S., Li, Y., Dollár, P., & Girshick, R. (2022). Masked autoencoders are scalable vision learners. In Proceedings of the IEEE/CVF Conference on Computer Vision and Pattern Recognition (pp. 16000-16009).
>
> [2] Chen, X., Xie, S., & He, K. (2021). An empirical study of training self-supervised vision transformers. In Proceedings of the IEEE/CVF International Conference on Computer Vision (pp. 9640-9649).
>
> [3] Manas, O., Lacoste, A., Giró-i-Nieto, X., Vazquez, D., & Rodriguez, P. (2021). Seasonal contrast: Unsupervised pre-training from uncurated remote sensing data. In Proceedings of the IEEE/CVF International Conference on Computer Vision (pp. 9414-9423).
>
> [4] Ayush, K., Uzkent, B., Meng, C., Tanmay, K., Burke, M., Lobell, D., & Ermon, S. (2021). Geography-aware self-supervised learning. In Proceedings of the IEEE/CVF International Conference on Computer Vision (pp. 10181-10190).
>
> [5] Caron, M., Touvron, H., Misra, I., Jégou, H., Mairal, J., Bojanowski, P., & Joulin, A. (2021). Emerging properties in self-supervised vision transformers. In Proceedings of the IEEE/CVF International Conference on Computer Vision (pp. 9650-9660).
>
> [6] Chen, T., Kornblith, S., Swersky, K., Norouzi, M., & Hinton, G. E. (2020). Big self-supervised models are strong semi-supervised learners. Advances in neural information processing systems, 33, 22243-22255.

---

### Official Review · Reviewer_T3bi · 2022-07-14

**Rating:** 4
**Confidence:** 3
**Soundness:** 3 good
**Presentation:** 3 good
**Contribution:** 2 fair

**Summary:**

In this paper, the authors proposed to apply the idea of masked autoencoder to satellite images. The authors proposed to directly handle temporal or spectral sequence of satellite images using a plain vision transformer model with extra encodings. Different masking strategies have been explored. The authors also repurposed existing datasets for multi-spectral satellite imagery.

**Questions:**

Please see the cons for details.

**Limitations:**

More discussion on the possible sub-optimal designs in the current method should have been provided.

**Strengths And Weaknesses:**

Pros:
1. The idea of masked autoencoder for satellite images is intuitive and straightforward.
2. Strong performance is shown in the empirical evaluations.
3. The paper is well presented.

Cons:
1. The main concern is on the insufficient ablation studies. The insights from existing ablation studies are actually limited.
a. Although the authors explored several techniques proposed for satellite images, the relative gains are actually smaller than using test-time augmentation from existing work.
b. From a subjective view, the distribution of pixels on satellite images is probably quite different from natural images. It is unclear whether the choice on mask ratio and the design of the decoder has similar impact on the final performance, which is actually the case even for MAE and VideoMAE. However, these important ablations that were explored in existing MAE works are not covered.
c. The selection of the size of the backbone is also limited, where only ViT-Large is used and it makes not directly comparable to existing results.
d. Moreover, justification on the choice of resolution, number of epochs and other hyper-parameters could be also of great value for future research.

2. An important baseline needs more explanation and exploration. It is not clear whether ImageNet Initialization is obtained from supervised pretraining or MAE. Actually both of them should have been evaluated to understand the contribution of training masked auto-encoder on the same domain of data.

---

> ### Author Response · Authors · 2022-08-02
> **Response to Reviewer T3bi**
>
>
> Thank you for your thoughtful feedback, recognition of SatMAE’s technical contributions and strong performance, and suggestions for improvement on more ablation studies and baseline. Your suggestions greatly helped us improve our analysis of SatMAE. You may find responses to your questions below.
>
> **Q:** *Relative gains of proposed techniques smaller than using test-time augmentation*
>
> **A:** In temporal data settings, test-time augmentation (TTA) is a natural add-on for improving model performance regardless of the model backbone. With a series of input images of the same location, averaging the model's predicted logits usually brings improvement. Our method exploits the temporal structure of data during training, but also benefits from test-time augmentation, though it is not our contribution. We would like to point out that the gain from our method without TTA against the strongest baseline would be to compare 76.78% (MAE ||, or second last row in Table. 2 in our *revised* table) with 79.69%, which is our method without TTA in Table. 4. This comparison shows a gain in accuracy of 2.91% using our method without TTA against a 1.8% gain in accuracy from including TTA. There is a significant improvement in our design over the already strong baseline of MAE.
>
> To be more convincing, we also performed another baseline by using standard MAE + TTA, which achieved 78.90% Top. 1 Acc. There is a 2.59% gap between this baseline and our method with TTA (81.49%). Judging from these numbers, we demonstrate that our method could benefit from TTA roughly as much as the baseline, even though we already exploited temporal information during training.
>
> **Q:** *Impact of mask ratio and decoder design on the final performance.*
>
> **A:** We have included further comparisons in our revised Appendix. Specifically, we compare a mask ratio of 0.6 and 0.9, in both the fMoW Sentinel and fMoW RGB temporal settings. We also test a patch size of 32 and 22 in the fMoW RGB temporal setting and a patch size of 16 in the fMoW Sentinel setting. Making the patch size smaller suffers from computational constraints, so we only tried larger patch sizes for ablation. The conclusion is that the current patch size outperforms larger patch sizes (intuitively), and the mask ratio of 0.75 is optimal among 0.6, 0.75, 0.9. This is expected since 0.75 is also the default mask ratio in MAE. We are surprised, though, that on temporal dataset our method is quite sensitive to the mask ratio. This may be a future research direction.
>
> We also further consider different spectral band grouping strategies for fMoW Sentinel. While we would have liked to explore more granular hyperparameter settings, we are constrained by the scale of available computational resources. As each experiment can take 3-7 days (training time varies under different patch sizes and mask ratios) pre-training plus an additional 1-2 days of finetuning even with 8 V100 GPUs, we hope the reviewer can understand our constraints in academic research labs.
>
> **Q:** *Only ViT-Large is used as the backbone.*
>
> **A:** In our revision, we include an ablation with ViT-Base in Table 5. We picked ViT-Large as a reasonable trade-off between expressivity and computational constraints for our experiments, as was done in the MAE paper. As we are constrained by computational resources, we only have the ability to focus on one model architecture setting.
> As shown the previous paper [1,2] ViT-Large constantly outperforms ViT-base, we thus make the assumption that ViT-Large would outperform ViT-Base. As we are targeting temporal and multi-spectral data, the memory consumption is already very high. Using ViT-Huge, though expected to push the performance even higher, requires a much longer training time or larger GPU memory.  We would like to also experiment with these variants in the future if we have more computational resources.  We believe our conclusion on ViT-large can also generalize to other ViT backbones.
>
> In the future, we will make pre-trained versions of the other ViT backbones available.
>
> **Q:** *The choice of resolution, number of epochs, and other hyper-parameters could be also of great value for future research.*
>
> **A:** We agree that this is a great future research direction. We have provided some details for the choice of our hyperparameters in Appendix section A3. Specifically, we used most of the default settings of the hyperparameters from existing works (e.g. MAE [1]) to establish the robustness of our method. Since varying these hyperparameters induces a large search space, we were constrained in many of our choices and therefore stuck to defaults where possible. We will leave a more in-depth analysis as future work.
>
> To be continued.

---

> ### Author Response · Authors · 2022-08-02
> **References Accompanying Our Response to Reviewer T3bi**
>
> **References:**
>
> [1] He, K., Chen, X., Xie, S., Li, Y., Dollár, P., & Girshick, R. (2022). Masked autoencoders are scalable vision learners. In Proceedings of the IEEE/CVF Conference on Computer Vision and Pattern Recognition (pp. 16000-16009).
>
> [2] Dosovitskiy, A., Beyer, L., Kolesnikov, A., Weissenborn, D., Zhai, X., Unterthiner, T., ... & Houlsby, N. (2020). An image is worth 16x16 words: Transformers for image recognition at scale. arXiv preprint arXiv:2010.11929.

---

> > ### Comment · Reviewer_T3bi · 2022-08-09
> > **Good response**
> >
> > Thanks for the response! Most of my concerns are addressed. Look forward to the final version.

---

> > > ### Author Response · Authors · 2022-08-09
> > > **Thank you for your response**
> > >
> > > Thank you very much for considering our revisions and we are glad that most of your concerns are addressed. Your constructive comments were very helpful in improving our paper. If the paper looks better to you, would you consider updating the numerical score?

---

> ### Author Response · Authors · 2022-08-02
> **Response to Reviewer T3bi (Cont.)**
>
> **Q:** *Clarification on ImageNet Initialization.*
>
> **A:** Thank you for this suggestion. We have clarified the ImageNet weights used in our experiments in the table captions in the *revised* paper. Specifically, we initialize the ViT-Large backbone with pre-trained weights on ImageNet (self-supervised setting) with standard MAE, as a fair comparison to our self-supervised learning method. Note that our method only pre-trained on fMoW RGB temporal/fMoW Sentinel, which was exposed to much less data than the ImageNet pre-trained setting. We have also added an experiment to finetune the ViT-Large with **supervised** ImageNet weights on fMoW Sentinel and fMoW RGB, which is supposed to be an even stronger baseline since it is exposed to ImageNet labels. In tables 1 and 3 of the revised paper, we see that the effect of using supervised ImageNet weights is muted compared to MAE weights. The results of the seventh row Table. 3 in our revised paper shows that this seemingly stronger setting actually performs worse probably due to the great difference in the distribution of ImageNet and fMoW-RGB/ fMoW Sentinel datasets.

---

### Official Review · Reviewer_FjEo · 2022-07-16

**Rating:** 7
**Confidence:** 4
**Soundness:** 3 good
**Presentation:** 3 good
**Contribution:** 3 good

**Summary:**

The paper presents a pre-training framework based on masked auto encoders for multispectral and multi-temporal satellite images.

**Questions:**

NA

**Limitations:**

Apart from poverty prediction, infrastructure development, and population estimation, most models for satellite imagery are developed to estimate environmental changes happening across the globe.

As you mentioned in the Conclusion - your model requires expensive computational resources. Although you also mention that you will be working in the future to make the model more efficient, if you could also comment on the carbon footprint generated from your model (or any future work that cites and uses your current model) vs its usability for assessing environmental impacts/climate change/green house gas emissions, that would be great! It would also add a bit broader impact, and trade-off analysis.

**Strengths And Weaknesses:**

The architecture proposed is novel as it addresses the temporal irregularities in satellite images, and builds a masked auto encoder for a variety of purposes such as image classification, segmentation etc. The method has been tested for a variety of benchmark datasets and the results are promising. The authors also acknowledge the large amount of computational resources that are required to build such an architecture. The paper is well written overall.

---

> ### Author Response · Authors · 2022-08-02
> **Response to Reviewer FjEo**
>
> Thank you for your thoughtful feedback, recognition of SatMAE’s novelty and contributions, and suggestions for improvement. We appreciate your insightful suggestion on elaborating more on the environmental impacts of our work. We provide a response below.
>
> **Q:** *Add comment on the carbon footprint generated from your model (or any future work that cites and uses your current model) vs its usability for assessing environmental impacts/climate change/green house gas emissions*.
>
> **A:** We include a brief analysis of the carbon footprint of training the model below.
>
> Our experiments were mainly conducted using Google Cloud Platform in region us-central1, which has a carbon efficiency of 0.57 kg $\text{CO}_2$eq. per kWh. For a model pre-trained and finetuned on fMoW RGB (temporal) dataset, a cumulative of 960 hours of computation was required on hardware of type Tesla V100-SXM2-16GB (TDP of 250W). Total emissions are estimated to be 136.8 kg $\text{CO}_2$eq. of which 100 percents were directly offset by the cloud provider. Estimations were conducted using the Machine Learning Impact calculator presented in [1]. For a model pre-trained and finetuned on fMoW Sentinel dataset, total emissions are estimated to be 109.44 kg $\text{CO}_2$eq. We list a table for the rough estimations down below:
>
> |  Experiment Setting    |   GPU hours (V100, us-central1)   |   Carbon Footprint (kg $\text{CO}_2$ eq.)   |
> | ---- | ---- | ---- |
> |  Pre-training SatMAE on fMoW RGB temporal |  768    |   109.44   |
> |   Finetuning SatMAE on fMoW RGB temporal   |   192   |   27.36   |
> |  Pre-training SatMAE on fMoW Sentinel |  576    |   82.08   |
> |   Finetuning SatMAE on fMoW Sentinel  |   192   |   27.36   |
> |   Finetuning SatMAE on NAIP  |   30   |   4.27   |
> |   Finetuning SatMAE on EuroSAT  |   4   |   0.57   |
> |   Finetuning SatMAE on SpaceNet V2  |   50   |   7.12   |
> |   Finetuning SatMAE on BigEarthNet 10%  |   16   |   2.28   |
>
> We acknowledge that training these models is computationally expensive. We hope that by making our pre-trained models publicly accessible, other researchers would not need to repeat the expensive pre-training steps and could instead simply finetune our model for their particular tasks. On fMoW RGB (temporal), finetuning only takes roughly 25% computational cost compared to pre-training. Smaller datasets are even less expensive, e.g. with SatMAE pre-trained on fMoW Sentinel, finetuning on EuroSAT only takes around 4 GPU hours but achieves a very high performance as shown in Table. 5 in our paper.
>
> As an effective SSL method, pretraining once could potentially save the cost of many downstream jobs of training from scratch by speeding up convergence significantly. We will release pre-trained weights on fMoW RGB temporal and fMoW Sentinel, and hope that they could benefit research in a variety of remote sensing tasks not only by pushing the performance but also by saving the computational costs.

---

> > ### Comment · Reviewer_FjEo · 2022-08-08
> > **Great response, thanks**
> >
> > Great response, thanks

---

> > > ### Author Response · Authors · 2022-08-09
> > > **Thank you for your response**
> > >
> > > Thanks again for the constructive feedback and recognition of our work!

---

> ### Author Response · Authors · 2022-08-02
> **References Accompanying Our Response to Reviewer FjEo**
>
> **References:**
>
> [1] Lacoste, A., Luccioni, A., Schmidt, V., & Dandres, T. (2019). Quantifying the carbon emissions of machine learning. arXiv preprint arXiv:1910.09700.

---

### Author Response · Authors · 2022-08-02
**Overall Author Response to Reviews**

We thank the reviewers for the constructive feedback! We are glad that the reviewers think our paper is well presented (R1, R2); our approach is intuitive, effective and promising (R1, R2, R3); our paper proposes a relatively novel setting or method (R1, R3, R4); our proposed method has been well tested with sufficient results (R1, R3). We are also glad that R3,  R4 acknowledge the impact of our collected dataset.

The main contribution of the paper is to present a novel self-supervised representation learning approach for pre-training and transfer learning on satellite imagery. While previous works have also attempted to do so, ours has a few advantages:
- It flexibly handles temporal and multi-spectral imagery
- It is conceptually intuitive and does not rely on designing special augmentations of the images to generate positive pairs
- It transfers very well to a host of downstream supervised learning tasks

The main questions and concerns raised by the reviewers are:
- The ablation study may not be comprehensive and convincing enough
- The baseline may not include all the state-of-the-art SSL methods or necessary variants
- Effectiveness of the different components of our design

In the revision, we have done the following to incorporate the reviewers’ feedback:
- Added extra discussion on the environmental impacts of our work in terms of its carbon footprint. (R1)
- Run extra ablation experiments on the spectral grouping design, the backbone, and the inclusion of the spectral embedding in Table 5 (R2, R3, R4).
- Run extra ablation experiments on patch size and mask ratio selection for both fMoW-temporal and fMoW RGB (in Appendix A.4 and A.5) (R2).
- Run extra baseline experiments of ViT-backbone MoCo V3 on fMoW-sentinel and SeCo on fMoW-temporal in tables 2 and 3 (R3).
- Clarify that we use both supervised and self-supervised ImageNet initializations in table 3 (R2).
- Clarify the writing of our encoding strategy with proper notation (R4).
- Include a discussion of relevant SITS literature in our Related Work section (R4).

We would like to emphasize SatMAE’s potential as a foundational model and strong baseline for multiple remote sensing tasks. As we mention in our conclusion, improving performance across such tasks “could enable rapid and accurate extraction of imagery features relevant for critical downstream tasks, including poverty prediction, infrastructure development, and population estimation.  Such information could aid governments in more rapid and data-informed decision making and ultimately bring large societal benefits.”

We also want to acknowledge that our experiments are computationally expensive. For instance, our method on fMoW RGB (temporal) dataset takes around 5 days for 8 V100 GPUs (16GB memory). As bottlenecked by computational resources, we have tried our best to perform the extra ablation studies requested by the reviewers. We believe we have presented all the crucial baselines required to demonstrate the effectiveness of our approach, and we hope the reviewers can understand. In the following, we address each reviewer’s questions one by one.

Link to the revision of the paper: https://openreview.net/references/pdf?id=CXZ2h_csDT9z. Additions are marked in blue for easier visibility.

Link to revised Appendix: https://openreview.net/attachment?id=WBhqzpF6KYH&name=supplementary_material.

---

### Meta-Review · Area_Chair_HpQK · 2022-08-26

**Recommendation:** Accept
**Confidence:** Certain

**Metareview:**

Four experts in the field reviewed the paper and recommended Accept, Borderline Reject, Weak Accept, and Weak Accept. According to the reviews, using MAE in satellite imagery is straightforward, but the novelty lies in details, such as using extra tokens to handle temporal consistency, masking strategies, etc. Another major question from the reviewers was about the ablation studies, and the rebuttal addressed it well. Hence, the decision is to recommend the paper for acceptance. We encourage the authors to consider the reviewers' comments and make the necessary changes to the best of their ability. We congratulate the authors on the acceptance of their paper!


**Award:**

No

---

### Decision · Program_Chairs · 2022-09-14

Accept